# Unconventional structure and mechanisms for membrane interaction and translocation of the NF-κB-targeting toxin AIP56

Johnny Lisboa [1,2] ✉, Cassilda Pereira [1,2], Rute D. Pinto[1], Inês S. Rodrigues [1,2], Liliana M. G. Pereira[1], Bruno Pinheiro [1,2,3], Pedro Oliveira[4], Pedro José Barbosa Pereira [5,6], Jorge E. Azevedo [7,8,9], Dominique Durand[10], Roland Benz [11], Ana do Vale [1,2] & Nuno M. S. dos Santos [1,2] ✉

Bacterial AB toxins are secreted key virulence factors that are internalized by target cells through receptor-mediated endocytosis, translocating their enzymatic domain to the cytosol from endosomes (short-trip) or the endoplasmic reticulum (long-trip). To accomplish this, bacterial AB toxins evolved a multidomain structure organized into either a single polypeptide chain or non-covalently associated polypeptide chains. The prototypical short-trip single-chain toxin is characterized by a receptor-binding domain that confers cellular specificity and a translocation domain responsible for pore formation whereby the catalytic domain translocates to the cytosol in an endosomal acidification-dependent way. In this work, the determination of the three-dimensional structure of AIP56 shows that, instead of a two-domain organization suggested by previous studies, AIP56 has three-domains: a non-LEE encoded effector C (NleC)-like catalytic domain associated with a small middle domain that contains the linker-peptide, followed by the receptor-binding domain. In contrast to prototypical single-chain AB toxins, AIP56 does not comprise a typical structurally complex translocation domain; instead, the elements involved in translocation are scattered across its domains. Thus, the catalytic domain contains a helical hairpin that serves as a molecular switch for triggering the conformational changes necessary for membrane insertion only upon endosomal acidification, whereas the middle and receptor-binding domains are required for pore formation.

Bacterial toxins are crucial virulence factors that lead to death or dysfunction of target cells, greatly contributing to the pathology in the host organism. Many toxins act on intracellular targets and are either injected into the cytosol by specialized protein secretion systems[1] or contain the components that allow them to reach the cytosol autonomously, as is the case with AB toxins[2,3]. Indeed, AB toxins have a modular multidomain structure typically organized in two distinct components: (i) component A, displaying enzymatic activity and targeting a crucial eukaryotic cytosolic factor; and (ii) component B, capable of binding to receptor(s) and conferring cellular specificity. The components can be encoded by one gene, originating a single-chain toxin, or by independent genes, resulting in multicomponent toxins with variable stoichiometry (AB$_2$, AB$_5$, A$_2$B$_5$, AB$_{7/8}$)[3,4]. AB toxins are internalized by the target cell through receptor-mediated endocytosis and their catalytic domains reach the cytosol of the cell either directly from the endosomal compartment upon acidic pH-triggered unfolding (short-trip toxins) or from

the endoplasmic reticulum (ER), after retrograde transport from the endosomal compartment to the Golgi and from there to the ER (long-trip toxins)[3–5]. In short-trip toxins, the B component usually includes a receptor-binding domain/region and a translocation domain/region, the latter responsible for the formation of a pore through which the catalytic component passes into the cytosol[3,4].

Of specific interest to this work are the short-trip single-chain toxins, whose prototypical domain organization is exemplified by diphtheria toxin (DT), with a catalytic domain-containing A-component (DTa) and a B component formed by a translocation domain and a receptor-binding domain[3,4,6–8]. The catalytic and translocation domains are normally linked by a disulfide bridge between cysteine residues located at the ends of a short peptide, creating a loop called linker peptide or simply linker[3,4]. These toxins are usually secreted in an inactive form, often becoming active upon proteolytic nicking at the linker by a bacterial or host protease. Proteolytic nicking divides the toxin into components A (light chain) and B (heavy chain), which are held together by the disulfide bridge[3,4]. After translocation, the reducing environment of the cytosol disrupts the disulfide bond, releasing the catalytic domain. Some toxins, such as Cytotoxic Necrotizing Factors (CNFs)[9,10], in which the disulfide bond is absent, contain a linker peptide between the A and B components that only becomes amenable to proteolytic cleavage after endosomal acidic pH-induced unfolding of the toxin. Moreover, there are other variations to the prototypical organization, as well as to the size and structural composition of the domains in some single-chain toxins, as exemplified by the large clostridial toxins, whose translocation and receptor binding is mediated by the same domain[11–15].

Apoptosis-inducing protein of 56 kDa (AIP56) is a major virulence factor of *Photobacterium damselae* subsp. *piscicida* (*Phdp*)[16], a Gram-negative bacterium that infects warm water marine fish with high economic importance for aquaculture[17]. AIP56 is a short-trip single-chain AB toxin[18,19] secreted by the type II secretion system of *Phdp*[20]. It has zinc-metalloprotease activity towards NF-kB p65, and targets fish phagocytes leading to their elimination by post-apoptotic secondary necrosis[16,18,21,22]. Ex-vivo experiments have shown that mouse macrophages also undergo NF-kB p65 inactivation and apoptosis following intoxication with AIP56[19], suggesting that the toxin recognizes a phylogenetically conserved, but yet unidentified, receptor(s). It has been previously shown that AIP56 is endocytosed by a clathrin- and dynamin-dependent process and undergoes acidic pH-dependent translocation from the endosomes to the cytosol, through a process assisted by Hsp90 and cyclophilin A/D[19,23].

In previous studies[18], amino acid sequence and limited proteolysis analyses suggested a two-domain organization for AIP56: (i) an A domain homologous to NleC (non-LEE encoded effector C), a type III effector with enzymatic activity towards NF-kB p65 that is injected into the cytosol by the type III secretion system of several enteric bacteria associated with human diseases[24–26]; and (ii) a receptor-binding B domain, homologous to Protein D from bacteriophage APSE2 (*Acyrthosiphon pisum* secondary endosymbiont 2)[27–30]. The two domains of AIP56 would be connected by an unusually long linker peptide (35 amino acids), the extremities of which are connected by a disulfide bridge required for intoxication[31]. However, contrary to most single-chain toxins[3,4], AIP56 does not require proteolytic cleavage (nicking) to become catalytically active. Importantly, nicking of its linker peptide abrogates toxicity, but not toxin internalization, suggesting that an intact linker is required for the translocation step[18].

Genes encoding AIP56-like toxins are also found in the genomes of *Candidatus* Symbiopectobacterium, *Arsenophonus nasoniae*, *Shewanella psychrophila* and of several *Vibrio* species, the latter including strains isolated from humans[32]. In addition, several genes encoding putative toxins containing a domain similar to either the catalytic domain or the receptor-binding domain of AIP56 (AIP56-related toxins) are also present in several prokaryotes and eukaryotes genomes[33].

In this work, the three-dimensional structure of AIP56 was determined using X-ray crystallography and the elements involved in acidic pH-induced conformational changes, membrane interaction and translocation were characterized.

## Results

### Three-dimensional structure of AIP56

The three-dimensional structure of AIP56 was determined at 2.5 Å resolution (data collection and refinement statistics are summarized in Supplementary Table 1). The crystallographic asymmetric unit contains four copies of AIP56 (chains A to D; root mean square deviation [rmsd] of 0.3–0.4 Å over 437–452 aligned Cα atoms), but the extent of the inter-protomer contacts is compatible with a monomeric organization, in agreement with small-angle X-ray scattering (SAXS) data (see below). The final crystallographic model of AIP56 (Fig. 1a) does not include residues 290-295 (ND1), 362-385 (ND2), 422-429 (ND3) and 454–455 (ND4), which are poorly defined in the electron density maps. The missing regions (ND1-4) were added with Modeller[34] based on the best model (out of 5) generated with AlphaFold2_advanced[35,36] (Supplementary Fig. 1a).

The three-dimensional structure of AIP56 (Fig. 1a and Supplementary Fig. 1a) shows a three-domain organization: (i) a N-terminal NleC-like catalytic domain (N1-F255) with a conserved zinc-binding motif (H165ExxH); (ii) a middle domain (G256-E307), with a disulfide bond (C262 and C298) flanking the 35 amino acid long linker peptide (S263 to E297), packed against the catalytic domain; and (iii) a C-terminal domain (P308-N497) previously shown to be involved in receptor binding[18].

SAXS analysis of AIP56 (Fig. 1b, c, d, Supplementary Table 2 and Supplementary Fig. 1b) revealed a reasonable fit between the experimental scattering data and the theoretical scattering profile calculated with CRYSOL[37] for the structural model of monomeric AIP56 obtained with Modeller ($\chi^2 = 5.05$) (Fig. 1b, upper panel), which is considerably worse when using the crystal structure of AIP56 without the missing linkers (ND1-4, $\chi^2 = 39$). The monomeric nature of AIP56 in solution is confirmed by the remarkable agreement between the sequence-derived molecular mass and experimental estimates (Supplementary Table 2 and Supplementary Fig. 1b). The corresponding residual plot showed non-random features that appeared to worsen at a $q$ range below 0.2 Å$^{-1}$ (Fig. 1b, bottom panel). Using normal mode analysis (NMA) with SREFLEX[38] it was possible to model the interdomain conformational flexibility of AIP56, therefore improving the overall fit with the experimental scattering data ($\chi^2 = 1.20$) and a much improved residual plot (Fig. 1c, d). The experimental AIP56 structure was compared with the structures of 19 putative AIP56-like toxins (Supplementary Table 3) predicted by AlphaFold2_advanced[35,36]. All structures are highly similar, with major differences observed only in the middle domains, particularly in the linker region (Supplementary Fig. 1c). Moreover, despite the high aminoacidic (Supplementary Fig. 2b) and structural (Supplementary Fig. 1c) conservation of the catalytic and receptor-binding domains, the AlphaFold-generated structures for *A. nasoniae* toxins resulted in distinct relative positions of those domains. Whether this results from inaccurate predictions of the middle domains by AlphaFold2 or corresponds to the real positions of the catalytic and receptor-binding domains in *A. nasoniae* toxins remains to be investigated.

While BLAST searches retrieve many genes encoding domains homologous to the catalytic and/or receptor-binding domains of AIP56, genes encoding domains homologous to just the middle domain are not found. Together with structural data and primary structure analysis (Supplementary Fig. 2), this suggests that AIP56 and AIP56-like toxins arose from the fusion of a gene homologous to *nleC*[24,25], encoding the catalytic domain, with a gene homologous to the one encoding Protein D from bacteriophage APSE2[27–29], which codes for the receptor-binding domain[18], with the middle domain likely

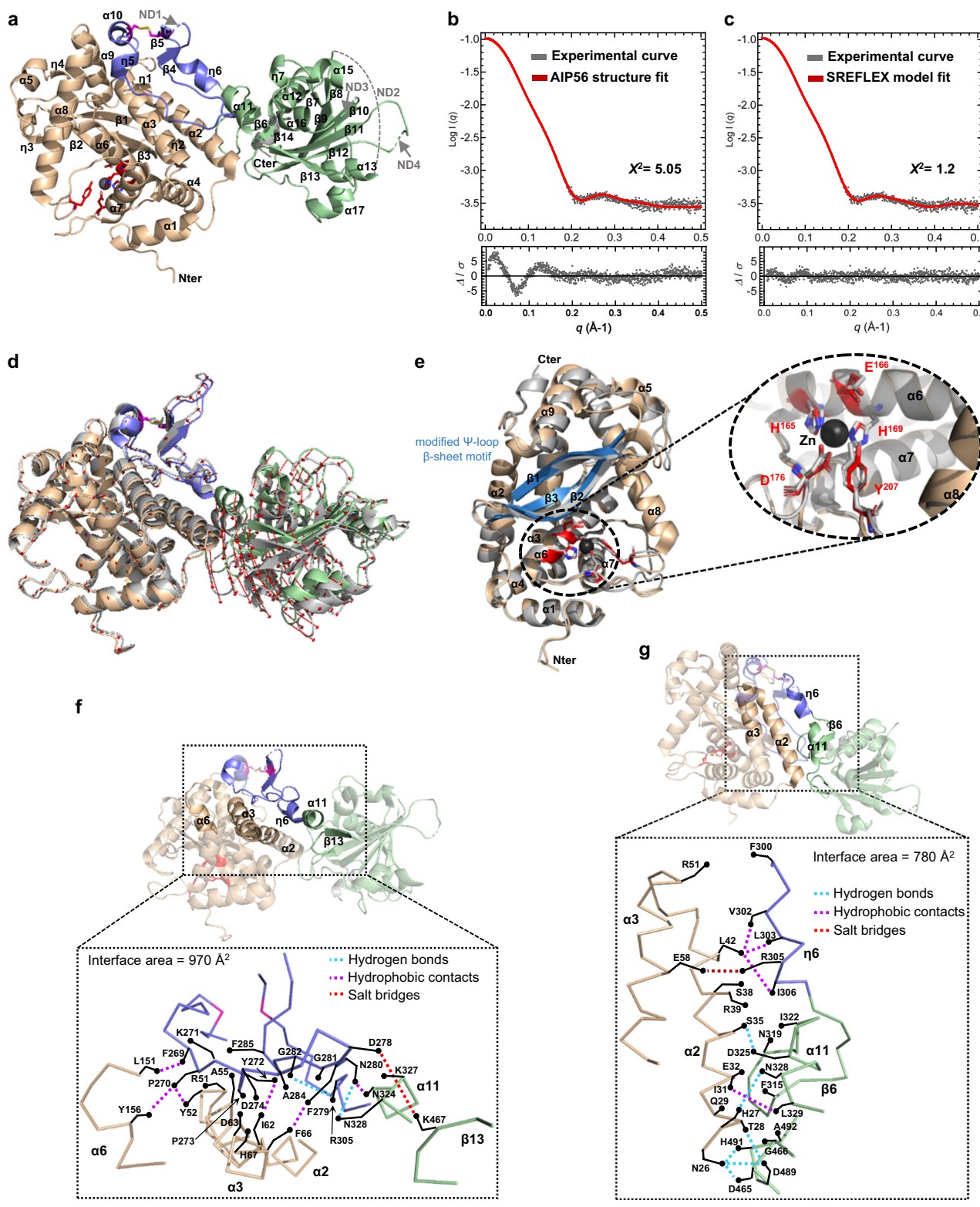

evolving from a gene/intergenic region of the recipient genome, where those genes were incorporated. Alternatively, the middle domain evolved also as an independent gene, but homologous genes are not yet available in the databases or are difficult to detect due to high sequence divergence.

### Detailed analysis of AIP56 structure

The catalytic domain of AIP56 has an overall Zincin fold similar to that of NleC (PDB entry 4Q3J; rmsd of 1.4 Å for 257 aligned Cα atoms), with

three β-strands forming a modified Ψ-loop β-sheet motif (β1-β3) overlaid on 9 α-helices (Fig. 1e)[24]. The active-site zinc ion of AIP56 is coordinated by H165 and H169 from helix α6 and by D176 located at the loop between helices α6 and α7. In two of the AIP56 monomers of the asymmetric unit, Y207 from the loop between helices α7 and α8 is the fourth zinc ligand, similar to what is observed in NleC[24]. The conserved E166 is ideally placed to promote the nucleophilic attack. As described before for NleC[24], the dimensions and shape of the active-site cleft of AIP56, as well as the pattern of glutamate and aspartate side

**Fig. 1 | Three-dimensional structure of AIP56. a** Cartoon representation of the AIP56 monomer (Chain A), with domains colored wheat (catalytic domain), blue (middle domain) or green (receptor-binding domain). The active site residues are shown as red sticks, the disulfide bond (C262 and C298) as magenta sticks and the zinc ion as a black sphere. The regions not defined (ND) in the structure are indicated by arrows or represented by a dashed line. The N- (Nter) and C-termini (C-ter) are labeled. **b** SAXS profile showing adjustment of the curve (red line) calculated from the structural model of AIP56 obtained by the Modeller program (Supplementary Fig. 1a), as calculated in CRYSOL, to the experimental scattering curve (gray dots) of AIP56. **c** SAXS profile showing better fitting of the SREFLEX model (red line). The bottom panels in (b) and (c) show the residual plots for the respective fits, with the residuals defined as $\Delta/\sigma = [\text{Iexp}(q) - \text{Icalc}(q)]/\sigma\text{exp}(q)$. $\chi^2$ scores were calculated in CRYSOL. **d** Complete SREFLEX simulated movement after the refinement stage. Vectors (red arrows) are drawn connecting the equivalent residues from the crystal structure (colored as in a) to the SREFLEX model (gray). **e** Cartoon representation of the superposed catalytic domains of AIP56 (wheat) and NleC (gray; PDB: 4Q3J). The modified Ψ-loop β-sheet motif (β1–β3) of AIP56 is colored blue. The N- (Nter) and C-termini (C-ter) and secondary structure elements are labeled. The inset shows a close-up of the active sites, including the zinc ion (Zn; black sphere) and the coordinating residues from AIP56 (red) and NleC (gray). **f** The linker peptide contacts both the catalytic and the receptor-binding domains. Top, cartoon representation highlighting the contact regions; Bottom, C-alpha trace of the contact region with contacts within a range of 4 Å is represented. Top and Bottom representations colored as in (**a**). **g** The catalytic and receptor-binding domains contact directly through helix α2 of the catalytic domain. Top and Bottom representations colored as in (**a**).

chains along the ridges of the cleft, mimic the DNA backbone phosphates to which NF-κB p65 binds[39–41] (Supplementary Fig. 3a). In NleC, four of the acidic residues in the cleft ridges - E115, E118, D139 and E150 - were shown to be important for efficient proteolysis[24]. Of these, E115 and E150, which are conserved in AIP56 (E96 and E131, respectively) (Supplementary Fig. 3a), are the most relevant[24]. This is probably also the case for AIP56 given that neither E118 nor D139 from NleC are conserved in AIP56 (the corresponding residues are N99 and N120, respectively) but, despite this, AIP56 cleaves human p65 in vitro more efficiently than NleC (Supplementary Fig. 3b).

The middle domain of AIP56 is composed of one α-helix (α10), two $3_{10}$ helices (η5 and η6) and two short antiparallel β-strands (β4-β5) (Fig. 1a and Supplementary Fig. 1a). Two cysteine residues, one in helix α10 and the other in strand β5, form a disulfide bond previously shown to be important for AIP56 toxicity[18,31]. These two cysteines flank the partially structured linker peptide that comprises helix η5 and strand β4 plus a connecting 15 residue-long unstructured loop (P270-A285) that contains an aspartate-rich patch at its tip (D274HDDD278) (Supplementary Fig. 2). The linker peptide establishes contacts with both the preceding catalytic and the succeeding receptor-binding domain (Fig. 1f).

AIP56's receptor-binding domain[18], comprises a central antiparallel seven-stranded β-sheet (β7–β13) sandwiched by helices α11, α12, α15 and α16 on one side, and helices α13, α14 and α17 on the other (Fig. 1a and Supplementary Fig. 1a). Two additional short β-strands (β6 and β14) form a small β-sheet perpendicular to the large one. Despite its simple architecture, the receptor-binding domain of AIP56 has an unusually high content (84 out of 190) of aromatic (11 tyrosine, 8 tryptophan and 14 phenylalanine) and hydrophobic (11 alanine, 9 isoleucine, 15 leucine and 16 glycine) residues. Although the central twisted antiparallel β-sheet of the AIP56's receptor-binding domain is similar to that of other unrelated proteins (Supplementary Fig. 3c), the 7 α-helices that surround the central β-sheet have a unique organization. Although this domain shows some flexibility with respect to the catalytic and middle domains (Fig. 1d), it establishes a series of contacts with helix α2 of the catalytic domain that prevent large structural movements (Fig. 1g).

## Pore formation requires both the middle and receptor-binding domains

Many short-trip single-chain toxins exhibit a three-domain organization, where the middle domain is dedicated to pore formation and facilitates the translocation of the catalytic domain into the cytosol[2,3]. This raises the question of whether the middle domain of AIP56, despite its structural simplicity, could mediate pore formation. The structural organization of AIP56, together with previous results showing that a chimera comprising β-lactamase (Bla) fused to the middle and receptor-binding domains of AIP56 (Bla$^{L19-W286}$AIP56$^{L258-N497}$) (Fig. 2a) was able to deliver Bla into the cytosol of mouse bone marrow-derived macrophages (mBMDM)[23] support this concept. To further test this, several truncated versions of AP56 were used in experiments

with black lipid bilayers (Fig. 2a), namely: the catalytic domain alone (AIP56$^{N1-G256}$), the catalytic domain with the middle domain (AIP56$^{N1-E307}$), the middle domain with the receptor-binding domain (AIP56$^{L258-N497}$) and the receptor-binding domain alone (AIP56$^{T299-N497}$). Of these, only the construct containing both the middle and receptor-binding domains (AIP56$^{L258-N497}$) displayed membrane-interacting activity after acidification (pH 4.8–5.0) of the cis-side of the membrane, although the observed activity was different and much lower than that obtained with intact AIP56 (Fig. 2b). In agreement, the Bla$^{L19-W286}$AIP56$^{L258-N497}$ chimera interacted with black lipid membranes, whereas a chimera with the AIP56 receptor-binding domain replaced by the DT receptor-binding (DTR) domain (AIP56$^{N1-E307}$DTR$^{S406-S560}$) did not display membrane activity (Fig. 2b) and was unable to deliver the AIP56 catalytic domain into the cytosol of U-2 OS cells (Supplementary Fig. 4a, b). Altogether, the black lipid bilayer experiments suggest that the interaction of AIP56 with the membrane requires both the middle and receptor-binding domains.

Of note, contrary to AIP56, the membrane-interacting activity of which is characterized by current bursts and "flickering" electrophysiological behavior[19] similar to that of TcdA and TcdB[14, 42,43] and requires strong acidification (pH 4.8-5.0)[19], Bla$^{L19-W286}$AIP56$^{L258-N497}$ formed stable pores at pH 6 (Fig. 2b). The finding that Bla$^{L19-W286}$AIP56$^{L258-N497}$ interacted with black lipid membranes at higher pH than AIP56 raised the question of whether the translocation of Bla to the cytosol depended on endosomal acidification. Contrary to AIP56 (Supplementary Fig. 4c)[19], translocation of Bla was not inhibited by Concanamycin A (Fig. 2c and Supplementary Fig. 4c, d), a vacuolar-type H$^+$-ATPase inhibitor that prevents endosomal acidification[44].

The difference in behavior between Bla$^{L19-W286}$AIP56$^{L258-N497}$ and the wild type toxin with respect to pH suggested that the AIP56 catalytic domain likely includes pH-responsive elements somehow required for its translocation in the context of full-length toxin.

## An amphipathic hairpin comprising helices α8-α9 of the catalytic domain is important for translocation

The pH-dependent membrane insertion in short-trip AB toxins depends on protonation of key titratable residues, generally located in the loop of an amphipathic helical hairpin of their translocation domains[9–11,13,45–52], although other protonatable residues outside the hairpin loop may also be involved[6,48,53–57]. Those residues are responsible for controlling pH-dependent conformational changes that lead to pore formation by the translocation domain[9,10,13,46–50,56,57].

At the C-terminal end of the catalytic domain of AIP56, a helical hairpin formed by two amphipathic helices (α8: D209 to H231 and α9: E234 to K247) (Fig. 1a and Supplementary Fig. 5a, b) contains residues potentially available for protonation (E214, E218, H222, H231 and E234) (Fig. 3a and Supplementary Fig. 5a, b). Except for H231, shielded by the middle domain in the crystal structure, all other residues are solvent exposed and, therefore, available for proton exchange (Fig. 3a). Indeed, the calculated electrostatic surface potential of the catalytic domain of AIP56 suggests that acidification modifies the surface

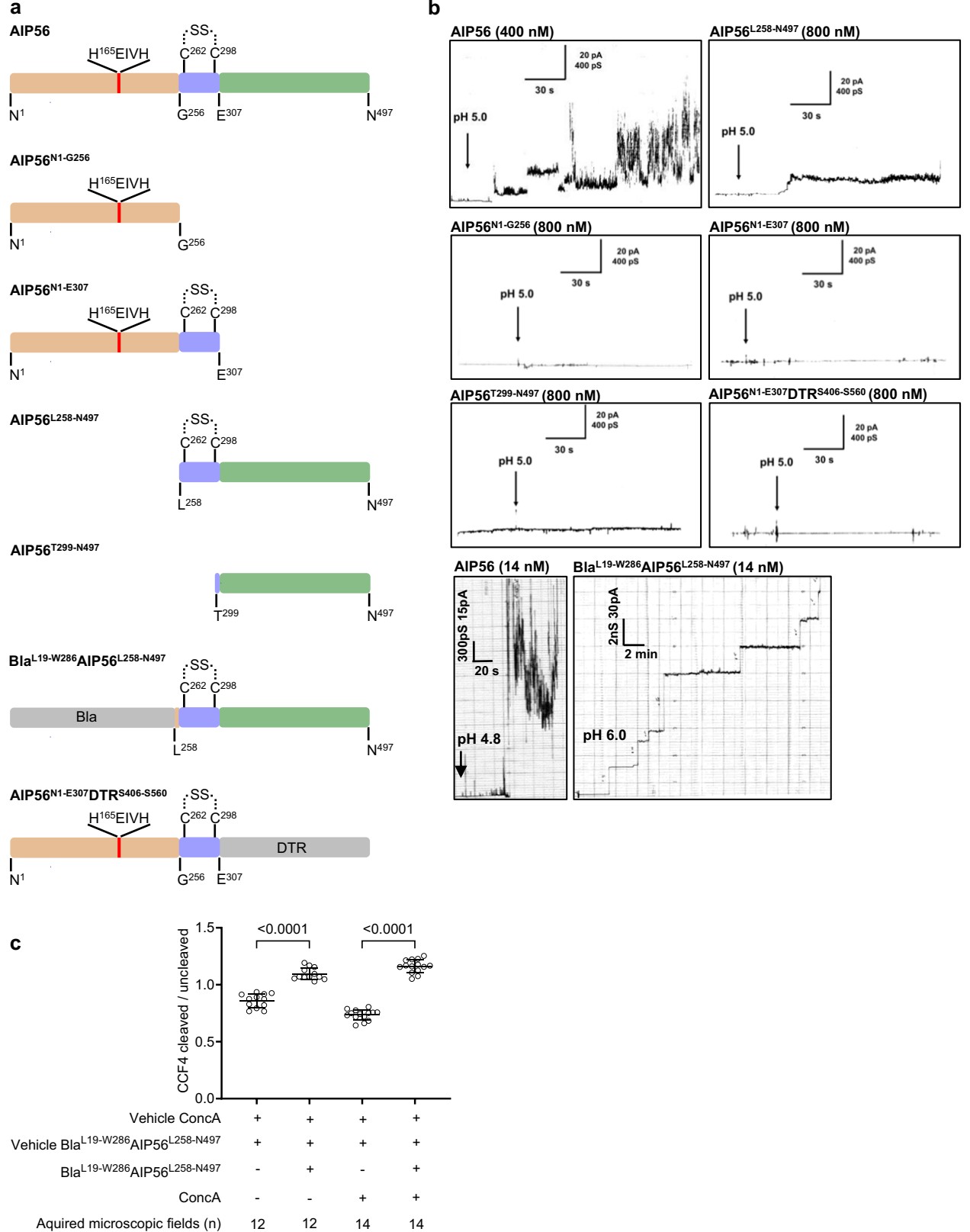

charge distribution in the putative pH-sensing region (Supplementary Fig. 5c). AIP56 single (E214K, E218K, H222K, H231K, and E234K) and multiple (H231K/E234K and E214K/E218K/H222K) residue mutant variants were assessed for cytotoxicity in mBMDM using AIP56-dependent NF-kB p65 cleavage as readout[19]. With the exception of AIP56$^{E214K}$, all other variants were non-toxic and unable to cleave p65

upon incubation with mBMDM (Fig. 3b), indicating that E218, H222, H231 and E234 are required for AIP56 toxicity. Importantly, the absence of toxicity did not result from incorrect folding (Supplementary Fig. 6a) or from the lack of catalytic activity of the variants (Supplementary Fig. 6b). Furthermore, the inactive variants blocked the endocytic uptake of AIP56 in competition assays (Supplementary

**Fig. 2 | Pore formation requires both the middle and receptor-binding domains. a** Schematic representation of AIP56 variants and chimeric proteins. AIP56 domains are colored as in Fig. 1a; β-lactamase (Bla) and diphtheria receptor-binding (DTR) domain, gray. **b** Only AIP56$^{L258-N497}$ and Bla$^{L19-W286}$AIP56$^{L258-N497}$ interacted with artificial black lipid membranes. Single-channel record of DiPhPC/n-decane membranes after addition of the indicated proteins to the cis-side of the black lipid bilayer at a final concentration indicated in the figure. Measurements were performed with 50 mV (AIP56$^{L258-N497}$) or 150 mV (Bla$^{L19-W286}$AIP56$^{L258-N497}$) at room temperature. Membrane activity was induced by acidification (pH 4.8–5.0; arrows) of the aqueous phase at the cis-side of the chamber with exception for Bla$^{L19-W286}$AIP56$^{L258-N497}$, which formed stable pores at pH 6. The average single-channel conductance was about 16 pS for 110 steps. Each result shown is representative of at least three ($n = 3$) independent measurements. **c** Pharmacological inhibition of vacuolar ATPase pump does not affect cytosolic delivery of Bla by Bla$^{L19-W286}$AIP56$^{L258-N497}$. The cleaved/uncleaved CCF4-AM ratios were determined by quantifying the indicated number of microscopic fields per condition. Representative images used for quantification are in Supplementary Fig. 4d. Results shown represent one out of three ($n = 3$) independent experiments. Statistical significance was tested by One-way ANOVA and $p$ values for individual comparisons were calculated by Tukey's HSD test and indicated on top of the brackets. Data are presented as mean values ± SD. Actual $p$ values from left-to-right: $p < 0.0001$, $p < 0.0001$. Data of the three independent experiments are provided in the Source data file. CCF4, Fluorescence Resonance Energy Transfer (FRET) substrate. ConcA concanamycin A.

Fig. 6c) and were efficiently endocytosed (Supplementary Fig. 6d), confirming that they retained the ability to interact with the cell surface receptor(s). These results suggest that the α8–α9 hairpin is important for the translocation process.

## Residues in the α8-α9 hairpin control the acidic pH-triggered conformational changes required for interaction with the membrane

To assess the potential impact of altering the acidic residues in the D209-K247 hairpin of AIP56 in the acidification-driven conformational changes of the toxin, the ability of the variants to bind 8-anilino-1-naphthalenesulfonic acid (ANS), a fluorophore that interacts with exposed hydrophobic regions in proteins[58,59], was assessed at different pH values (Fig. 3c and Supplementary Fig. 7a). In line with the results above, AIP56$^{E214K}$ displayed a behavior similar to that of AIP56 at all pH values tested. The non-toxic variants were more prone to change conformation in response to acidification, suggesting that E218, H222, H231, and E234 are involved in controlling acidic pH-triggered conformational changes of AIP56. Replacing E218 by a lysine had the least impact, whereas a maximal effect was observed with AIP56$^{H231K/E234K}$ and AIP56$^{H231K}$, followed by AIP56$^{E234K}$ (Fig. 3c). This suggested that H231, seconded by E234, plays a preponderant role in controlling the conformational changes induced by acidification. Accordingly, AIP56$^{H231K/E234K}$ is much more susceptible to proteolysis upon acidification than the native toxin (Fig. 3d and Supplementary Fig. 7b). However, after cleavage at the linker[18], the catalytic and receptor-binding domains of the native and variant toxins (Fig. 3d, bands A and B, respectively) are resistant to proteolysis, indicating that their globular state is maintained at least for some time at acidic pH. On the other hand, even considering some loss of efficiency of proteinase K and chymotrypsin at acidic pH, the native toxin is less prone to cleavage, suggesting that the linker region underwent structural changes with acidification that hindered its cleavage. Although in the crystal structure H231 is shielded from the solvent by the middle domain (Fig. 3a), the involvement of that residue in the pH-induced conformational alterations (Fig. 3c and Supplementary Fig. 7a) suggests flexibility of that domain or its linker peptide under physiologic conditions. This idea is further supported by the finding that at pH 8, chymotrypsin cleaved AIP56 (Supplementary Fig. 7b), between F285 and F286[18], a position that is not accessible in the crystal structure (Supplementary Fig. 7c). Taken together, the ANS and limited proteolysis results suggest that at acidic pH there is an increase in the general flexibility of the toxin, probably due to the disruption of contacts between the different domains (Fig. 1f, g), and a restructuring of the linker region.

The endosome acidification-driven conformational changes occurring in several short-trip AB toxins are known to be required for achieving a membrane-competent state that allows membrane interaction and insertion[60]. The effect of the AIP56 hairpin mutations in low pH-triggered membrane interaction was evaluated by analyzing the ability of the variants to interact with artificial black lipid membranes at acidic pH. In agreement with the results obtained in mBMDM

intoxication and ANS fluorescence assays, AIP56$^{E214K}$ displayed a membrane-interacting activity similar to that of native AIP56, whereas the remaining variants failed to interact with the lipid membranes upon acidification (Fig. 3e). Noteworthy, while a single replacement of the pH-sensing residues within the hairpin loop did not alter the pH-dependent conformational changes in DT[47,56] or cytotoxicity in TcdB[13], changing any of the pH-sensing residues of AIP56 abolished cytotoxicity and resulted in marked conformational changes, which were particularly drastic at pH <6 (Fig. 3c and Supplementary Fig. 7a). This suggests that, although AIP56 needs to undergo acidic pH-induced conformational changes for inserting into the membrane, these changes need to be controlled to avoid excessive unfolding and instability in solution, as shown previously for DT[53,54]. Such mechanism would also explain the inability of the non-toxic variants to insert in artificial membranes at low pH.

Notably, with the exception of the AIP56-like protein encoded by the longest *A. nasoniae* open reading frame (WP_051297127.1), all other homologous proteins have a conserved histidine equivalent to H231 of AIP56 (Supplementary Fig. 7d). Residues E218 and E234 are also highly conserved, but there is no conservation of H222, and E214 is conserved in only a few molecules.

Taken together, these results suggest that the hairpin within the catalytic domain of AIP56 contains pH-sensing residues that control the conformational changes necessary for pore formation by the middle and receptor-binding domains, a mechanism that may be conserved in AIP56-like toxins.

## Assistance of AIP56 translocation by Hsp90 requires the hairpin region

A Cytosolic Translocation Factor (CTF) complex, including Hsp90, thioredoxin reductase and coat protein complex I (COPI), has been described as required for efficient intoxication by several short-trip toxins[23,61–72]. A peptide motif (T1) in the translocation domain helix 1 (TH1) of DT, also present in the translocation domain of several other toxins (Supplementary Table 4), has been shown to assist DTa translocation by interacting directly with β-COP[73], one of the components of the CTF complex required for the arrival of DTa at the cytosol[61]. Studies with the multicomponent anthrax toxin have also uncovered the involvement of a T1-like motif, located in the catalytic components lethal or edema factors, in the translocation process[74,75], and confirmed its interaction with β-COP[76].

Translocation of AIP56 is also assisted by cytosolic factors, including Hsp90 and cyclophilin A/D, with the interaction occurring preferably when the toxin is unfolded, in line with the hypothesis that Hsp90 and cyclophilin A/D interact with AIP56 during or immediately after translocation[23]. Nevertheless, while both toxin domains interact with cyclophilin A/D in a similar way, the interaction with Hsp90 occurs preferentially through the catalytic domain (upstream of L258) of the toxin[23], although the interaction region has not been mapped in detail. Here, a T1-like peptide (A211RVEAIQERD) was identified in the first helix (D209 to H231) of the hairpin of AIP56's catalytic domain (Supplementary Fig. 5a and Supplementary Table 4), raising the possibility

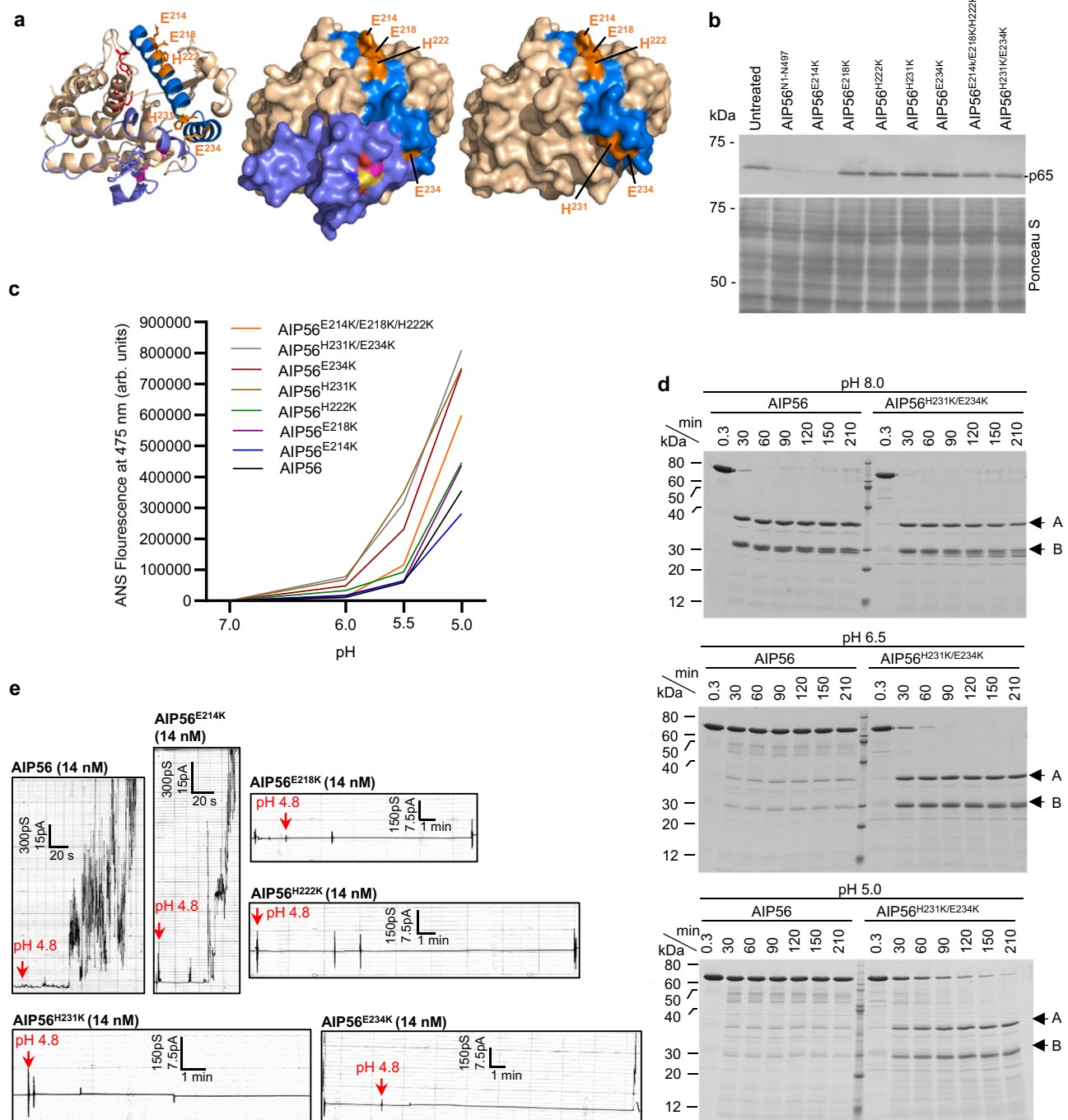

**Fig. 3 | Residues E218, H222, H231 and E234 in the D209-K247 hairpin control the low pH-triggered conformational changes required for AIP56 membrane interaction and translocation. a** Localization of the putative pH-sensing residues selected for replacement on the three-dimensional structure of the AIP56 catalytic domain. Cartoon (left) and surface representation of AIP56 catalytic domain with (middle) or without (right) the middle domain. The catalytic residues (H165, E166 and H169) are shown in red, the putative pH-sensing residues in orange and the D209-K247 hairpin in marine blue. The cysteine residues (pink sticks) forming the disulfide bridge (yellow) are also shown. **b** Analysis of NF-kB p65 cleavage in mouse bone marrow-derived macrophages (mBMDM) by V5 plus His-tagged AIP56 variants. Cleavage of p65 was assessed by western blotting (upper panel; chromogenic detection) and protein loading by staining the membranes with Ponceau S (lower panel). The result shown is representative of six (*n* = 6) independent experiments. **c** Peak ANS (8-Anilino-1-naphthalene-sulfonic acid) fluorescence measured at 475 nm for the indicated pH, normalized by subtracting the corresponding values at pH 7 (fluorescence due to

conformational changes caused by the mutation and not due to acidification). The measurement curves for each pH at different wavelengths are shown in Supplementary Fig. 7a. The results shown are representative of at least three (*n* = 3) independent experiments. AIP56, black; AIP56^E214K, blue; AIP56^E218K, purple; AIP56^H222K, green; AIP56^H231K, brown; AIP56^E234K, pink; AIP56^H231K/E234K, gray; AIP56^E214K/E218K/H222K, orange; **d** Coomassie Blue-stained SDS-PAGE gels from limited proteolysis of AIP56 and AIP56^H231K/E234K by Proteinase K. A and B mark the bands corresponding to the catalytic and receptor-binding domains, respectively. The results shown are representative of two (*n* = 2) independent experiments. **e** Interaction of AIP56 or AIP56 variants with black lipid bilayers. Single-channel recordings of DiPhPC/n-decane membranes after addition of the indicated proteins to one side of the black lipid bilayer at a final concentration of 14 nM. Membrane activity was induced by acidification (pH 4.8; red arrows) of the aqueous phase at the cis-side of the chamber. Each result shown is representative of at least three (*n* = 3) independent measurements. Source data for (**b**), (**c**) and (**d**) are provided in the Source data file.

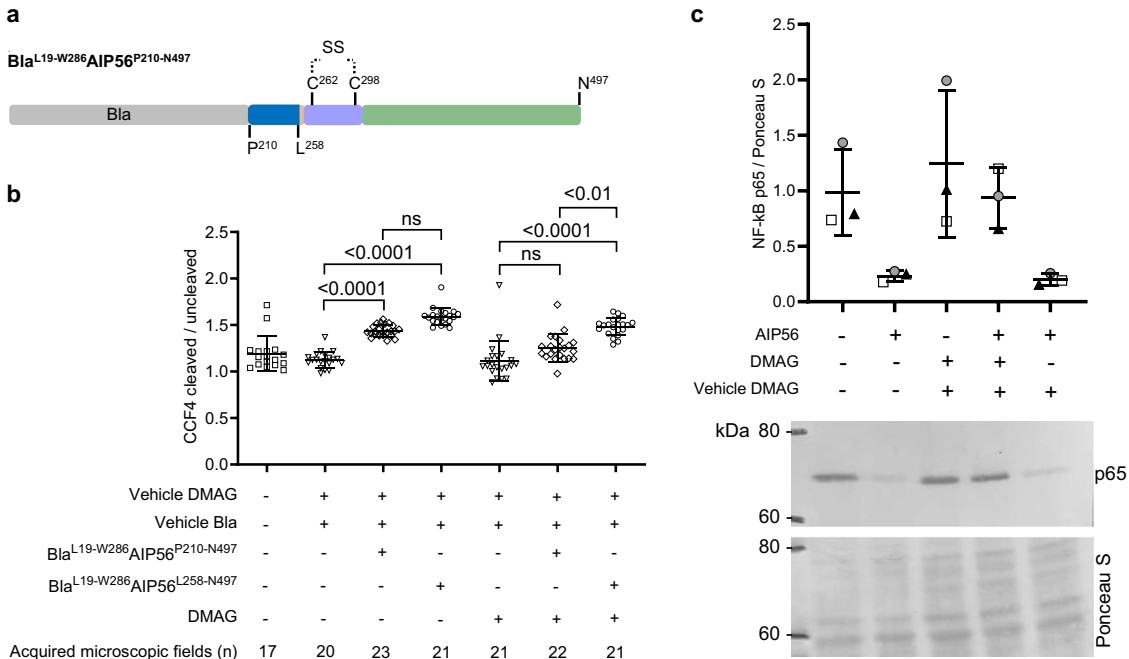

**Fig. 4 | Pharmacological inhibition of Hsp90 impairs the cytosolic delivery of Bla by Bla^L19-W286^AIP56^P210-N497^ but not of Bla^L19-W286^AIP56^L258-N497^. a** Schematic representation of chimera Bla^L19-W286^AIP56^P210-N497^. Bla, β-lactamase. **b** FRET-based assay to access the effect of Hsp90 inhibition on Bla delivery. The cleaved/uncleaved CCF4-AM ratios were determined by quantifying the indicated number of microscopic fields per condition. Results shown represent one out of three (*n* = 3) independent experiments. Statistical significance was tested by Kruskal–Wallis nonparametric test and the adjusted *p* values for individual comparisons were obtained by Bonferroni correction. Data are presented as mean values ± SD. Actual *p* values from left-to-right and upwards: *p* < 0.0001, *p* < 0.0001, *p* = 0.0993, *p* = 0.8533, *p* < 0.0001, *p* = 0.0051; ns non-significant, CCF4

Fluorescence Resonance Energy Transfer (FRET) substrate, DMAG 17-(dimethyla-minoethylamino)−17-demethoxygeldanamycin (Hsp90 inhibitor), Bla β-lactamase. **c** Control of 17-DMAG activity by confirming its inhibitory effect on NF-kB p65 (nuclear factor kappa-light-chain-enhancer of activated B cells subunit p65) cleavage upon AIP56 intoxication of mBMDM. A representative blot of three (*n* = 3) independent experiments is shown. Loading correction was achieved by dividing the density of p65 by the respective density of the Ponceau S staining. Data are presented as mean values ± SD. Different symbols represent independent experiments. DMAG 17-(dimethylaminoethylamino)−17-demethoxygeldanamycin (Hsp90 inhibitor). Source data for (**b**) and (**c**) are provided in the Source data file.

that this peptide or the region around it[73–77] mediates the interaction with Hsp90. To test this, a chimera comprising β-lactamase fused to a C-terminal portion of AIP56 containing the hairpin and the middle and receptor-binding domains (Bla^L19-W286^AIP56^P210-N497^) (Fig. 4a) was used in cellular assays. Like the chimera Bla^L19-W286^AIP56^L258-N497^ (without hairpin), Bla^L19-W286^AIP56^P210-N497^ (with hairpin) was able to translocate Bla to the cytosol of mBMDM (Fig. 4b). However, whereas delivery of Bla by Bla^L19-W286^AIP56^L258-N497^ is not prevented by the Hsp90 inhibitor 17-DMAG (Fig. 4b, c)[23], the translocation of Bla by Bla^L19-W286^AIP56^P210-N497^ is abolished upon Hsp90 inhibition (Fig. 4b, c), showing that translocation of Bla via this chimera is dependent on Hsp90, and suggesting that the hairpin region mediates the AIP56-Hsp90 interaction during translocation.

## An aspartate-rich patch in the linker is required for translocation but dispensable for pore formation

Contrary to many other known AB toxins, AIP56 does not require pre-activation to display in vitro enzymatic activity, but needs an intact linker peptide for toxicity[18,19]. The fact that previous studies showed that versions of AIP56 with a disrupted linker retain catalytic activity and interact with cell surface receptor(s), but yet do not cleave intracellular p65, suggest that the linker is involved in the translocation process[18,19]. Analysis of the amino acid sequences from the linker region of AIP56 homologs revealed a conserved aspartate-rich patch (Supplementary Fig. 2), suggesting a putative conserved function. To assess whether this patch of aspartate residues plays a role in the intoxication process, they were changed to serine (AIP56^D274S/D276-278S^) or asparagine (AIP56^D274N/D276-278N^) residues and the toxicity of the variants ascertained by evaluating p65 cleavage in mBMDM. Although all

variants were correctly folded (Supplementary Fig. 6a), retained catalytic activity (Supplementary Fig. 6b) and receptor-interaction capacity (Supplementary Fig. 6c, e), none of them was able to intoxicate mBMDM (Fig. 5a), suggesting that the conserved aspartate residues are required for AIP56 translocation. This hypothesis was further tested by assessing the impact of replacing the aspartate residues in the ability of the toxin to translocate directly from the cell membrane into the cytosol in response to an acidic pulse, using a previously optimized translocation assay[19]. The results showed that the translocation of both AIP56^D274S/D276-278S^ and AIP56^D274N/D276-278N^ across the host cell membrane in response to an acidic pulse was impaired (Fig. 5b). The exact function of the aspartate-rich fragment is currently unknown, but is probably not related to membrane interaction or insertion, since the variants interacted with artificial lipid membranes (Fig. 5c). These results reinforce previous evidence that the linker peptide of AIP56 is involved in the translocation process[18,19].

## Discussion

AIP56 is the only AB-type toxin characterized to date that targets NF-kB[18]. It has a general intoxication mechanism[19] similar to other short-trip single-chain AB toxins that after receptor-mediated endocytosis undergo acidic pH-dependent conformational changes resulting in the exposure of hydrophobic regions, followed by membrane interaction with pore formation and translocation of the catalytic domain from endosomes into the cytosol[3, 4]. The mechanism of translocation is well described for several of those toxins including DT[7,46–48,50,56,73], clostridial neurotoxins[78–82], *Pasteurella multocida* toxin[49] and CNFs[9,10,83], all containing a dedicated middle translocation domain[9,10,45–50] or, as in large clostridial toxins, a common translocation and receptor-binding

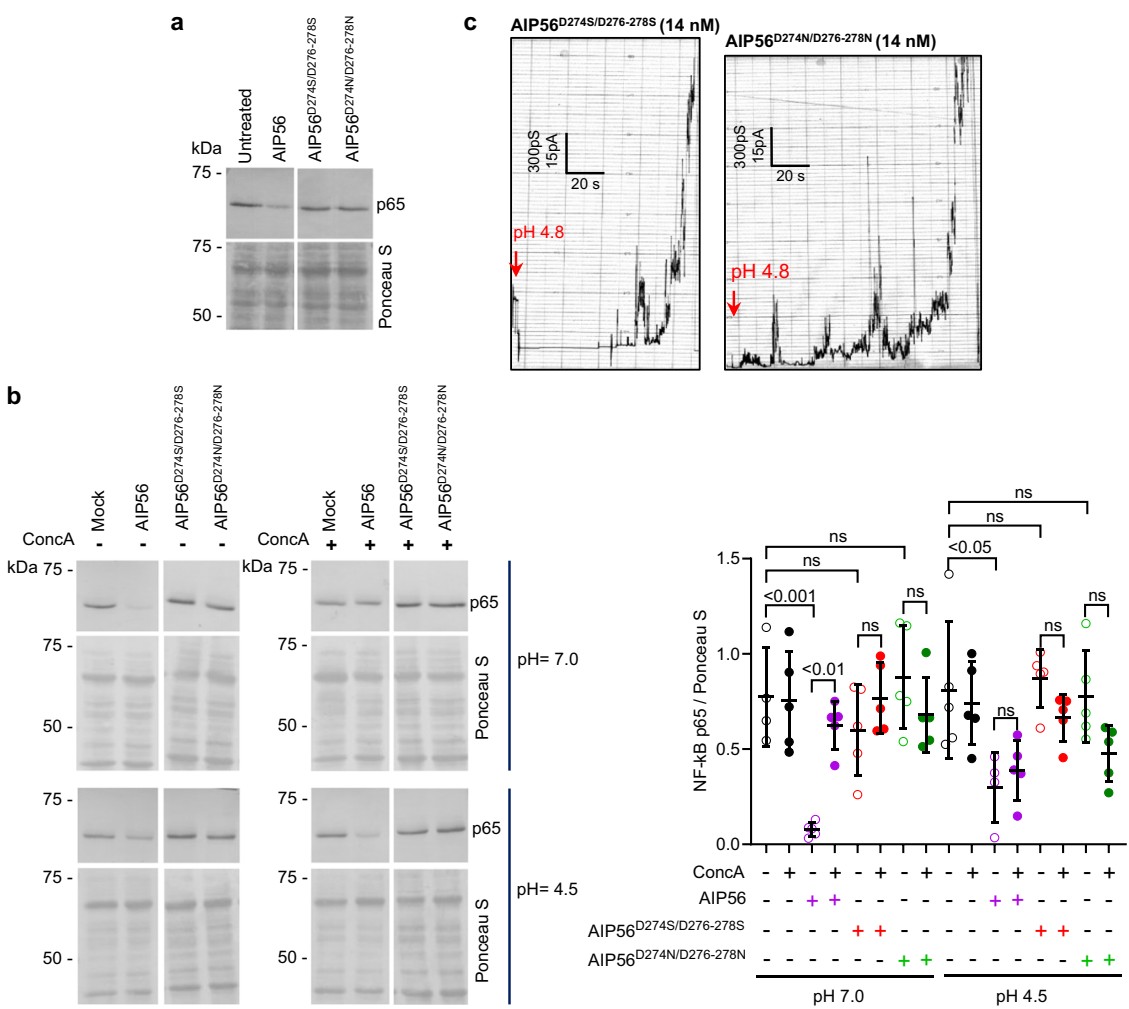

**Fig. 5 | An aspartate-rich patch located in the linker is important for AIP56 translocation but not for pore formation. a** V5 plus His-tagged AIP56 modified in the aspartate-rich motif (AIP56D274S/D276-278S and AIP56D274N/D276-278N) is unable to cleave p65 in intact cells. Cleavage of p65 was assessed by western blotting and protein loading by Ponceau S staining. The result shown is representative of six (n = 6) independent experiments. **b** V5 plus His-tagged AIP56D274S/D276-278S and AIP56D274N/D276-278N were unable to translocate across the host cell membrane in response to acidification. In all experiments, mock-treated cells were used as controls. NF-kB p65 cleavage was analyzed by western blotting. The result shown is representative of five (n = 5) independent experiments. The plot shows the quantification of intact NF-kB p65 normalized for Ponceau S. Statistical significance was tested by one-way ANOVA and p values for the individual comparisons were calculated using Tukey's HSD test. Data are presented as mean values ± SD. Actual p values from left-to-right and upwards for pH 7.0: p < 0.001,

p = 0.91, p > 0.99, p = 0.005, p = 0.90, p = 0.81; for pH 4.5: p = 0.02, p > 0.99, p > 0.99, p > 0.99, p = 0.78, p = 0.36; ns non-significant. Open (without concanamycin A) or closed (with concanamycin A) symbols as well as color coding have been added to facilitate the reading of the experimental conditions, as specified below the graph. Samples were derived from the same experiment and the blots processed in parallel. NF-kB p65, nuclear factor kappa-light-chain-enhancer of activated B cells subunit p65; ConcA, concanamycin A (black); AIP56, purple, AIP56D274S/D276-278S, red; AIP56D274N/D276-278N, green. **c** AIP56D274S/D276-278S and AIP56D274N/D276-278N retained the ability to interact with black lipid bilayers. Proteins were used at a final concentration of 14 nM. Membrane activity was induced by acidification (pH 4.8; red arrows) of the aqueous phase at the cis-side of the chamber. Each result shown is representative of at least three (n = 3) independent measurements. Source data for (**b**) and (**c**) are provided in the Source data file.

domain[14]. However, AIP56 lacks a domain functionally and structurally equivalent to the translocation domain of those toxins. Instead, the small size and simple structure of the AIP56 middle domain apparently resulted in the spread of the structural elements involved in the AIP56 translocation process across all of its domains.

Indeed, in AIP56, the pH-sensing residues that control the conformational changes required for triggering membrane interaction and translocation seem to be located at the carboxyl-terminal portion of the catalytic domain, in a helical hairpin that is not involved in pore formation. Thus, it is proposed that the pH-sensing residues prevent pore formation until the low pH conditions found in the endosome induce unfolding of the catalytic domain for translocation through the pore. Accordingly, the BlaL19-W286AIP56L258-N497 chimera (without hairpin) could not only form stable pores in the artificial lipid bilayers at a

higher pH than AIP56 but also translocate Bla to the cytosol even when endosomal acidification was prevented by Concanamycin A.

The results presented here also show that the middle and receptor-binding domains are both required for pore formation since: (i) interaction with black lipid bilayers was only observed with constructs that have both domains (AIP56L258-N497 and BlaL19-W286AIP56L258-N497); (ii) the middle and receptor-binding domains of AIP56 were sufficient to deliver Bla into the cytosol (BlaL19-W286AIP56L258-N497); and (iii) disruption of the linker region within the middle domain[18] or replacing the AIP56 receptor-binding domain by the DT receptor-binding domain (AIP56N1-E307DTRS406-S560) abrogates cell toxicity, as monitored by p65 cleavage. The specific role each of these domains plays in pore formation requires further study. However, it is plausible that a region(s) within the receptor-binding domain promotes toxin oligomerization

and the middle domain forms the transmembrane channel. Indeed, the small size and lack of structural complexity of the middle domain accords with the channel-forming regions that in β-PFTs[84–88] and anthrax PA[51,52] refold to a β-hairpin that participates in the formation of a β-barrel, being particularly similar to the aerolysin insertion loop (prestem loop or tongue)[86,88] (Supplementary Fig. 8a). While in β-PFTs[84–88] and PA[51,52] the conformational alterations of the insertion regions is structurally restricted, with release and refolding to β-hairpin triggered by proteolytic- and/or pH-dependent activation, the linker region of AIP56 appears to be relatively flexible at physiological pH.

Given the observed resistance of both the catalytic and receptor-binding domains of AIP56 to proteolytic cleavage, it can be proposed that after binding to its receptor(s) and endocytosis, acidification will trigger a structural alteration of the pH-sensing hairpin leading to increased overall flexibility of the toxin and reshaping the linker region into a β-hairpin which will be part of a β-barrel channel formed upon toxin oligomerization. In this scenario, although the disulfide bridge is required for the intoxication process of AIP56[31], it is not clear whether it plays a role in pore formation or is simply required for maintaining the catalytic and receptor-binding domains close together. It should be noted that the involvement of a disulfide bridge on the release of the catalytic domain in the cytosolic reducing environment is not mandatory, as supported by the fact that CNFs lack a disulfide-bridged linker-peptide loop, but still contains a linker peptide between the catalytic and translocation domains that is cleaved after endosomal acidic pH-induced unfolding of the toxin[9,10]. In fact, in the short-trip single-chain toxins that contain a disulfide-bridged linker-peptide loop that requires cleavage, the linker peptide is short and located between the catalytic and translocation domains[3,4], whereas in AIP56 the linker peptide is much longer, making up almost entirely the middle domain. Furthermore, its cleavage abrogates translocation but not toxin internalization[18], supporting that it will be involved in the translocation process and may not be cleaved. Interestingly, the predicted structure of AIP56-related toxins exhibits a three-domain organization, including a known (e.g., CdtB-like domain) or predicted (e.g., presence of a zinc-metalloprotease binding motif) enzymatic domain, a structurally complex middle domain and a domain homologous to the receptor-binding domain of AIP56 (Supplementary Fig. 8b). This three-domain organization resembles that of many short-trip single-chain toxins, such as DT[7], clostridial neurotoxins[78–82], *P. multocida* toxin[49] and *Pseudomonas aeruginosa* exotoxin A[89], wherein their middle pore-forming/translocation domain is flanked by the catalytic and receptor-binding domains, which suggests that the middle domain in AIP56-related toxins and AIP56/AIP56-like toxins may be responsible for the formation of the transmembrane pore.

Finally, it is possible that upon translocating to the cytosol, the catalytic domain interacts with Hsp90 through the region of its pH-sensing hairpin[19,23].

In summary, the structural and functional data described in this work provide new insights on AIP56 translocation mechanism and may contribute to develop prophylaxis and treatments based on AIP56. Future work may elucidate whether the conclusions drawn for AIP56 can be extended to the increasing number of AIP56-like and -related toxins.

## Methods

### Ethics statement
This study was carried out in accordance with European and Portuguese legislation for the use of animals for scientific purposes (Directive 2010/63/EU; Decreto-Lei 113/2013). The work was approved by the ORBEA (Animal Welfare and Ethics Body) of i3S and was licensed by Direcção-Geral de Alimentação e Veterinária (DGAV), the Portuguese authority for animal protection (ref. 004933).

### Mice
C57BL/6J mice were bred and housed (filter top cages, eurostandard type II, corn cob bedding material and nesting paper and plastic tube rolls for enrichment) at the animal facility of the Instituto de Investigação e Inovação em Saúde with 12 h light/12 h dark light cycle, 20–24 °C and 45–65% humidity. The mice were fed sterilized food (2014S, Envigo) and water ad libitum and were euthanized by $CO_2$ inhalation followed by cervical dislocation.

### Reagents and antibodies
PMSF (P7626), pronase E from *Streptomyces griseus* (P5147), diphtheria toxin (DT) from *Corynebacterium diphtheriae* (D0564), Proteinase K from *Engyodontium album* (P2308), α-chymotrysin from bovine pancreas type II (C4129), 8-Anilino-1-naphthalenesulfonic acid (ANS, A1028) and concanamycin A (ConcA, C9705) were purchased from Sigma Aldrich. HBSS, HEPES, L-glutamine, sodium pyruvate and 17-(dimethylaminoethylamino)−17-demethoxygeldanamycin (17-DMAG, ant-dgl-5) were purchased from Invitrogen. DMEM (10938-025), heat-inactivated fetal bovine serum (FBS, 10500064) and penicillin/streptomycin (P/S, 15140122), all Gibco, and LiveBLAzer™ FRET-B/G Loading Kit with CCF4-AM (K1095) were purchased from Life Technologies. CCF4-AM is a lipophilic, esterified form of the CCF4 substrate, which is a Fluorescence Resonance Energy Transfer (FRET) substrate that consists of a cephalosporin core linking 7-hydroxycoumarin to fluorescein. Isopropyl β-D-1-thiogalactopyranoside (IPTG, MB026) was purchased from NZYTech. The anti-human NF-κB p65 C-terminal domain (clone c-20) rabbit polyclonal antibody (sc-372, dilution 1:3000) was from Santa Cruz Biotechnology and the anti-V5 (R960-25, dilution 1:5000) mouse monoclonal antibody was purchased from Invitrogen. Goat anti-IgG rabbit alkaline phosphatase conjugated secondary antibody (A9919, dilution 1:10000) and goat anti-IgG mouse alkaline phosphatase conjugated secondary antibody (A2429, dilution 1:10000) were purchased from Sigma Aldrich.

### Cells
Mouse bone marrow-derived macrophages (mBMDM) were derived from bone marrow of femurs and tibias from 4–8 week-old C57BL/6J male mice, as previously described[90]. Briefly, femurs and tibias were flushed with 10 mL of Hanks' balanced salt solution (HBSS) and the cells recovered by centrifugation and resuspended in supplemented DMEM (DMEM with 10 mM glutamine, 10 mM HEPES, 1 mM sodium pyruvate, 10% (v/v) FBS and 1% penicillin/streptomycin) plus 10% (v/v) of L929 cell conditioned medium (LCCM) as a source of macrophage colony stimulating factor[91]. Fibroblasts were removed by culturing the cells overnight, at 37 °C in a humidified chamber in a 7% (v/v) $CO_2$ atmosphere, on a cell culture dish. The non-adherent cells were collected, centrifuged, resuspended in supplemented DMEM plus 10% (v/v) LCCM at a concentration of $5 \times 10^5$ cells mL$^{-1}$, plated in 24-well cell culture plates at a density of $5 \times 10^5$ cells/well and incubated as above. Three days later, 10% (v/v) LCCM was added and on the 7th day, the medium was renewed. Cells were used at day 10. To obtain LCCM, L929 cells were grown in 75 cm$^2$ filtered cap flasks in supplemented DMEM until reaching 100% confluence. Cells were then diluted 1:100 in fresh supplemented DMEM and incubated for 10 days at 37 °C, 7% (v/v) $CO_2$. The supernatant was collected, pooled, centrifuged at 750 × $g$ for 10 min and filtered. LCCM was aliquoted and stored at −20 °C until used.

### DNA Constructs for recombinant protein production
Constructs encoding full-length AIP56 cloned into the NcoI/XhoI restriction sites of pET28a (Novagen) in frame with a C-terminal 6His-tag (pET28AIP56H+) or with a V5 plus 6xHis-tag (pET28AIP56V5H+) are described in ref. [16] and ref. [19], respectively. The coding regions of AIP56$^{N1-G256}$, AIP56$^{N1-E307}$, AIP56$^{L258-N497}$ and AIP56$^{T299-N497}$ were PCR-amplified using pET28AIP56H+ as template and cloned into the NcoI/

XhoI restriction sites of pET28a in frame with a 6xHis-tag at the C-terminus (AIP56$^{N1-G256}$, AIP56$^{N1-E307}$ and AIP56$^{T299-N497}$) or N-terminus (AIP56$^{L258-N497}$).

To obtain the constructs for expressing Bla-AIP56 chimeric proteins, the sequence encoding Bla$^{L19-W286}$ was amplified from plasmid p327 (gifted by Dr. Dimitri Panagiotis Papatheodorou) and cloned into pET28a NcoI/SacI restriction sites, yielding plasmid pET28Bla$^{L19-W286}$. DNA sequences encoding AIP56$^{P210-N497}$ or AIP56$^{L258-N497}$ were then amplified using pET28AIP56H+ as template and ligated into the SacI/XhoI restriction sites of pET28Bla$^{L19-W286}$ in frame with a C-terminal 6xHis-tag.

To generate chimera AIP56$^{N1-E307}$DTR$^{S406-S560}$, AIP56$^{N1-E307}$ encoding sequence was amplified from pET28AIP56H+ and cloned into pET28 NcoI/SacI restriction sites, yielding plasmid pET28AIP56$^{N1-E307}$. DTR$^{S406-S560}$ coding sequence (accession number WP_072564851.1) was then amplified from pET-22b DT 51E/148K (gifted by Dr. John Collier; Addgene plasmid # 11081; RRID:Addgene_11081) and ligated into the SacI/XhoI restriction sites of pET28AIP56$^{N1-E307}$ in frame with a C-terminal 6xHis-tag.

Constructs coding for AIP56$^{E214K}$, AIP56$^{E218K}$, AIP56$^{H222K}$, AIP56$^{H231K}$, AIP56$^{E234K}$, AIP56$^{E214K/E218K/H222K}$, AIP56$^{H231K/E234K}$, AIP56$^{D274S/D276-278S}$ and AIP56$^{D274N/D276-278N}$ with or without V5-tag were generated with the QuickChange Site-Directed Mutagenesis Kit (Stratagene, 200518) following the manufacturer's instructions, using pET28AIP56V5H+[19] and pET28AIP56H+[16] as template, respectively.

Plasmids and primers used in this study are listed in Supplementary Table 5.

## Production of recombinant proteins

Full-length His-tagged AIP56 (AIP56), full-length V5 plus His-tagged AIP56 (AIP56-V5), AIP56$^{N1-G256}$, AIP56$^{N1-E307}$DTR$^{S406-S560}$ and AIP56 mutant variants were expressed in *Escherichia coli* BL21(DE3), AIP56$^{N1-E307}$ in *E. coli* BL21-CodonPlus (DE3) and AIP56$^{T299-N497}$ in *E. coli* SoluBL21 (DE3). Bla-AIP56 chimeras and AIP56$^{L258-N497}$ were expressed in *E. coli* Rosetta (DE3). Transformed *E. coli* cells were cultured at 37 °C in 1 L of Luria Bertani (LB) broth with shaking (200 rpm), except for AIP56$^{N1-E307}$DTR$^{S406-S560}$ for which 6 L were used. At OD$_{600}$ ~ 0.6, 0.5 mM IPTG was added and protein expression carried out at 17 °C for 4 h in the case of Bla-AIP56 chimeras or for 20 h for all other proteins. Cells were harvested by centrifugation and resuspended in 30 mL of Buffer: (i) 50 mM Bis-Tris pH 6.5, 300 mM NaCl for AIP56, AIP56$^{N1-G256}$, AIP56$^{T299-N497}$ and AIP56 mutant variants; (ii) 50 mM Tris pH 8.0, 300 mM NaCl for AIP56$^{L258-N497}$, AIP56$^{N1-E307}$ and AIP56$^{N1-E307}$DTR$^{S406-S560}$; or (iii) 20 mM Tris pH 8.0, 200 mM NaCl, 5% (v/v) glycerol for Bla-AIP56 chimeras. In all cases, cell lysis was performed by sonication. After centrifugation, the recombinant proteins were purified from the supernatant using nickel-affinity chromatography (Ni-NTA agarose, ABT) followed by size-exclusion chromatography (Superose 12 10/300 GL, GE Healthcare), except in the cases of AIP56$^{N1-E307}$ and AIP56$^{N1-E307}$DTR in which only nickel-affinity chromatography was performed. Recombinant proteins were analyzed by SDS-PAGE and purities determined by densitometry of Coomassie Blue-stained gels (Image Lab Software, BioRad). Protein batches used in this work were ≥90% pure.

## Determination of protein concentration

The concentrations of recombinant proteins were determined by measuring absorbance at 280 nm using a NanoDrop 1000 and/or NanoDrop One (Thermo Fisher Scientific) considering the extinction coefficient and the molecular weight calculated with the ProtParam tool (http://www.expasy.org/tools/protparam.html).

## Crystallization

Initial crystallization hits for AIP56 were identified by high-throughput screening at the HTX Lab of the EMBL Grenoble Outstation (Grenoble, France). AIP56 crystals were obtained by mixing equal volumes of protein (7 mg mL$^{-1}$) and crystallization solution (400 mM sodium acetate, 100 mM Tris pH 8.5, 15% (w/v) PEG 4 K, 10 mM taurine), which was optimized using an additive screen kit (HR2-428; Hampton Research). Prior to data collection, crystals were successively transferred into the crystallization solution supplemented with 15% and 30% (v/v) glycerol, followed by flash-cooling in liquid nitrogen.

## Structure determination

Several X-ray diffraction datasets were collected from AIP56 crystals at Synchrotron SOLEIL (Saint-Aubin, France; beamlines Proxima-1 and Proxima-2) and at the European Synchrotron Radiation Facility (ESRF, Grenoble, France; beamlines ID23, ID29 and ID30). The best dataset was collected on beamline ID29 on a Pilatus 6 M (Dectris) detector (2700 images, 0.1° rotation, 0.02 s exposure, wavelength 0.972 Å), indexed and integrated with XDS[92] and analyzed using the STARANISO server as diffraction was highly anisotropic[93]. Statistics for data processing are summarized in Supplementary Table 1. The structure of AIP56 was solved by molecular replacement with PHASER[94] using an AIP56 model generated by the artificial intelligence program AlphaFold2_advanced[35,36]. Several cycles of model building and refinement were performed using COOT, Phenix.refine and BUSTER[95–99]. All illustrations of macromolecular models were produced with PyMOL[100]. The final refined coordinates and structure factors were deposited at the Protein Data Bank under PDB entry 7ZPF and the corresponding diffraction images at the SBGrid Data Bank (https://doi.org/10.15785/SBGRID/911).

## Small-angle X-ray scattering (SAXS) data acquisition, processing and analysis

X-ray scattering data were collected at the SWING beamline of the SOLEIL Synchrotron (Saint-Aubin, France). Measurements were performed using a HPLC Bio Sec3 (Agilent) size-exclusion column, online with the SAXS measuring cell, a 1.5 mm diameter quartz capillary contained in an evacuated vessel. The sample-to-detector (Dectris Eiger 4 M) distance was set to 2000 mm and the wavelength λ to 1.0 Å, allowing useful data collection over the momentum transfer range of 0.005 Å$^{-1}$ < q < 0.5 Å$^{-1}$ ($q = 4\pi \sin(\theta)/\lambda$). SAXS data were collected directly after elution of the AIP56 protein through the HPLC column equilibrated in 50 mM HEPES buffer pH 7.5, 500 mM NaCl. Fifty μl of protein sample were injected at 15 °C and two different initial concentrations, 13.4 and 3.2 mg mL$^{-1}$, in order to obtain a curve with a good statistic and devoid of interparticle effects. The flow rate was 0.2 mL min$^{-1}$, frame duration was 0.99 sec and the dead time between frames was 0.01 sec. Scattering of the elution buffer before void volume was recorded and used as buffer scattering for subtraction from all protein patterns. The scattered intensities were displayed on an absolute scale using the scattering by water. Data were first analyzed using Foxtrot and then using the US-SOMO HPLC module[101]. This program provided for each SAXS frame the values of the scattering intensity I(0) and of the radius of gyration Rg by applying the Guinier analysis together with a calculation of the approximate molar mass using the Rambo and Tainer approach[102]. Identical frames under the main elution peak were selected using Cormap[103] and averaged for further analysis. Both averaged curves corresponding to the two initial concentrations were merged. Data was deposit in the Small-Angle Scattering Biological Data Bank under accession code SASDNW6.

## Intoxication assays

mBMDM monolayers seeded in 24-well plates (5 × 10$^5$ cells/well) were intoxicated by continuous incubation with 174 nM AIP56 or AIP56 variants, diluted in culture medium (supplemented DMEM + 10% (v/v) LCCM) at 37 °C for 4 h. The NF-κB p65 level was evaluated by western blotting after 2 h incubation with the toxin and was used as readout for arrival of AIP56 into the cytosol.

## Fluorescence Resonance Energy Transfer (FRET) based assay

mBMDM cells cultured at a density of $1.5 \times 10^5$ cells/well in 8-well plates (ibidi) were pre-treated with concanamycin A (ConcA; 10 nM), 17-DMAG (20 μM) or their vehicles (DMSO for ConcA and water for 17-DMAG) in supplemented DMEM without FBS for 1 h at 37 °C. Stock solutions (250 μM ConcA in DMSO and 8.1 mM 17-DMAG in water) were diluted with culture medium prior to the experiment. Next, the medium was replaced by fresh medium containing the inhibitors and Bla$^{L19\text{-}W286}$AIP56$^{P210\text{-}N497}$ or Bla$^{L19\text{-}W286}$AIP56$^{L258\text{-}N497}$ was added at a final concentration of 525 nM. Untreated cells and cells treated with the chimeras' vehicle were used as controls. The cells were incubated for 15 min at 37 °C, washed twice with PBS and loaded with CCF4-AM (LiveBLAzer™ FRET-B/G Loading Kit, Life Technologies; K1095), according to the manufacturer's instructions, in HBSS supplemented with 10 mM glutamine, 10 mM HEPES, 1 mM sodium pyruvate and 10% (v/v) FBS for 30 min at room temperature (RT). Subsequently, mBMDM were washed twice with supplemented HBSS and fixed at RT for 15 min in 4% (w/v) paraformaldehyde in Dulbecco's Phosphate Buffered Saline (DPBS). Cells were then observed with a CFI PL APO LAMBDA 40X/0.95 objective on a Nikon Eclipse Ti-E microscope (Nikon). The samples were illuminated at 395 nm by a SpectraX light engine (Lumencor) using a quad dichroic filter 310DA/FI/TR/CY5-A and emission filters 450/50 and 525/50 (Semrock). Images were acquired with an EMCCD camera iXon ULTRA 888 (Andor Technologies). In each experiment, a determined number (as provided in the respective figure) of microscopic fields per condition were acquired and 3 independent experiments were performed. Ratiometric analysis of the microscopic field images acquired for emission wavelengths (Em) 447 nm and 520 nm, corresponding to the ratio of cleaved CCF4-AM/intact CCF4-AM, was made using a custom-made ImageJ macro (https://github.com/PaulaSampaio/ALM-i3S_macros) on Fiji software45 (ImageJ version 1.51n, NIH, USA)[104]. In parallel to every FRET-based assay, the activity of ConcA or 17-DMAG was confirmed by evaluating the inhibitory effect of the drugs on AIP56 intoxication, as previously described[23]. For this, mBMDM were pre-treated with ConcA (10 nM), 17-DMAG (20 μM) or their respective vehicles, in supplemented DMEM without FBS for 1 h at 37 °C prior to incubation with 174 nM AIP56 for 2 h while maintaining the inhibitory conditions. Mock-treated cells, cells treated only with the toxins and cells treated only with inhibitors were used as controls. The cleavage NF-κB p65 was evaluated by western blotting and was used as readout of AIP56 toxicity.

## SDS-PAGE and western blotting

SDS-PAGE was performed using the Laemmli discontinuous buffer system[105]. Prior to loading, the samples were boiled for 5 min in SDS-PAGE sample buffer (50 mM Tris-HCl pH 8.8, 2% (v/v) SDS, 0.05% (v/v) bromophenol blue, 10% (v/v) glycerol, 2 mM EDTA, 100 mM DTT). For western blotting, the proteins were transferred onto Amersham™ Protran 0.45 μm nitrocellulose membranes (GE Healthcare Life Science). The protein loading on the membranes was controlled by Ponceau S staining. The membranes were blocked for 1 h at RT with 5% (w/v) skimmed milk in Tris-buffered saline (TBS) containing 0.1% (v/v) Tween 20 (TBS-T) followed by incubation for 1 h at RT with the primary antibodies diluted in blocking buffer. Immunoreactive bands were detected with alkaline phosphatase conjugated secondary antibodies and NBT/BCIP (Promega). Blots shown correspond to representative results of at least 2 independent experiments. Uncropped scans of all blots are provided in the Source Data file. The quantification of the blots was performed by densitometry using the Fiji software[104]. The results are expressed as the density of the p65 band relative to the density of the Ponceau S of the same lane. Each graph combines the results of at least two independent experiments.

## Interaction with black lipid bilayers

Analysis of the interaction of AIP56, AIP56$^{N1\text{-}G256}$, AIP56$^{N1\text{-}E307}$, AIP56$^{L258\text{-}N497}$, AIP56$^{T299\text{-}N497}$, AIP56$^{E214K}$, AIP56$^{E218K}$, AIP56$^{H222K}$, AIP56$^{H231K}$, AIP56$^{E234K}$, AIP56$^{D274S/D276\text{-}278S}$, AIP56$^{D274N/D276\text{-}278N}$, Bla$^{L19\text{-}W286}$AIP56$^{P210\text{-}N497}$, Bla$^{L19\text{-}W286}$AIP56$^{L258\text{-}N497}$ and AIP56$^{N1\text{-}E307}$DTR$^{S406\text{-}S560}$ with black lipid bilayers was performed as described previously[19,31,106]. Briefly, membranes were formed between two aqueous compartments of a Teflon chamber from DiPhPC/n-decane (diphytanoyl phosphatidylcholine [Avanti Polar Lipids, Alabaster, AL] in n-decane) in a 150 mM KCl solution buffered with 10 mM HEPES to pH 7.0 or 10 mM MES to pH 6.0. After the membrane had turned black, purified protein samples mixed 1:1 with cholesterol suspension in water were added to the cis compartment of the chamber while stirring. Single-channel recordings of the membrane current in the presence of recombinant protein were performed with 50 mV or 150 mV at RT. With exception of the Bla-AIP56 chimeras, all tested samples did not show any membrane activity under these conditions for at least 10 min, showing that the samples were essentially free of bacterial contaminants that could form channels (porins). For Bla chimeras, to inhibit eventual porin activity, 50 μM of norfloxacin (OmpF) and 50 μM of ceftobiprole (OmpC) were added in the cis-side of the chamber. When indicated, membrane activity was induced by acidification (pH 4.8–5.0) of the aqueous phase at the cis-side of the chamber. The membrane current was measured with a pair of Ag/AgCl electrodes with salt bridges switched in series with a voltage source and a highly sensitive current amplifier (Keithley 427). The amplified signal was recorded by a strip chart recorder. Each experiment was repeated at least three times.

## Limited proteolysis

AIP56 or AIP56$^{H231K/E234K}$ (5.36 μM) were incubated with Proteinase K (8.65 nM) on ice in 20 mM Tris pH 8.0, 200 mM NaCl or 20 mM Bis-Tris pH 6.5 or pH 5.0 with 200 mM NaCl. Aliquots were removed at different incubation times and the reaction stopped by adding SDS-PAGE sample buffer followed by heating at 95 °C for 5 min. Digests were analyzed by SDS-PAGE.

## pH-induced translocation assay

Translocation across the cell membrane in response to acidification was assessed using a protocol first described for diphtheria toxin[107,108] and used before for wild type AIP56[19,31]. Briefly, mBMDM were incubated for 30 min on ice with 174 nM AIP56 or AIP56 variants, in the absence or presence of 10 nM ConcA (that was maintained during the entire assay to inhibit normal toxin uptake). After removing the supernatant, the cells were incubated for 1 h at 37 °C with buffer at pH 4.5 or 7.0 followed by incubation in culture medium at pH 7.4 for 2 h. Different pH values were obtained by adding $H_3PO_4$ to a solution containing 0.5 mM $MgCl_2$, 0.9 mM $CaCl_2$, 2.7 mM KCl, 1.5 mM $KH_2PO_4$, 3.2 mM $Na_2HPO_4$ and 137 mM NaCl. In all experiments, mock-treated cells were used as controls. NF-kB p65 cleavage was analyzed by western blotting.

## Fluorescence assay with 8-Anilino-1-naphthalenesulfonic acid (ANS)

Binding of ANS to the recombinant proteins was measured by following the increase of the fluorescence signal (from 400 to 550 nm) on a Horiba Fluoromax-4 spectrofluorimeter (excitation at 380 nm). V5 plus His-tagged proteins (1.5 μM) were titrated into a quartz cuvette with 75 μM ANS. The experiments were conducted at 25 °C at different pH (100 mM ammonium acetate, 150 mM NaCl for pH 5.0 and 5.5; 100 mM MOPS, 150 mM NaCl for pH 6.0, and 100 mM HEPES, 150 mM NaCl for pH 7.0).

## Bioinformatic tools

The 10 amino acid T1-like motif in AIP56 was first identified in silico by Blasting (https://blast.ncbi.nlm.nih.gov/Blast.cgi) the known T1-like

motif from diphtheria toxin, botulinum neurotoxins serotypes A, C and D, *Clostridium difficile* toxin B (TcdB), cytotoxic necrotizing factor 1 (CNF1), *P. multocida* toxin and anthrax edema and lethal factors (Supplementary Table 1) against AIP56 sequence. The identified AIP56 T1-like motif was then analyzed by the Multiple Expectation maximization for Motif Elucidation MEME tool (https://meme-suite.org/meme/tools/meme)[109]. The helical wheel (Supplementary Fig. 5b) was drawn using Galaxy (https://cpt.tamu.edu/galaxy-pub)[110]. Surface charge distribution of the catalytic domain of AIP56 (Supplementary Fig. 5c) at pH 7.4, 6.0 and 5.0 was calculated using PDB2PQR and APBS[111] based on $pK_a$ values predicted by PROPKA3.1[112].

## Statistical analysis

In each legend, the size of the experiment is indicated. Normality of the data were assessed through the Shapiro–Wilk test. Except for Fig. 4b, one-way ANOVA was used to assess the differences between conditions. Regarding Figs. 2c and 5b, multiple comparisons were evaluated using the Tukey's HSD (honest significant difference) test. Since the data in Fig. 4b did not pass the normality test, differences between conditions were assessed by Kruskal–Wallis nonparametric test and the adjusted *p* values for individual comparisons were obtained by Bonferroni correction. Statistical significance was set to $p < 0.05$. All statistical analysis was performed with a confidence interval of 95%. Statistical analysis was performed using the IBM SPSS Statistics (v25) software (Armonk, NY). All graphs were constructed using GraphPad Prism 8 (GraphPad Software, San Diego, CA).

## Reporting summary

Further information on research design is available in the Nature Portfolio Reporting Summary linked to this article.

## Data availability

The final refined coordinates and structure factors generated in this study have been deposited in the Protein Data Bank under PDB entry 7ZPF. The corresponding diffraction images have been deposited at the SBGrid Data Bank (https://doi.org/10.15785/SBGRID/911). The SAXS data generated in this study have been deposited in the Small-Angle Scattering Biological Data Bank under accession code SASDNW6. The crystallography data of NleC used in this study are available in the Protein Data Bank under PDB entry 4Q3J. Source data are provided with this paper.

## Code availability

Custom-made ImageJ macro on Fiji software45 (ImageJ version 1.51n, NIH, USA)[104] is available at https://github.com/PaulaSampaio/ALM-i3S_macros.

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

## Acknowledgements

This work was supported by National funds through FCT under the project UIDB/04293/2020 and by FEDER funds through Programa Operacional Factores de Competitividade – COMPETE and by national funds through FCT – Fundação para a Ciência e a Tecnologia under the project PTDC/BIA-MIC/29910/2017 to N.M.S.S. A.d.V. was funded by Portuguese national funds through the FCT and, when eligible, by COMPETE 2020 FEDER funds, under the Scientific Employment Stimulus–Individual Call 2021.02251.CEECIND/CP1663/CT0016. We acknowledge access to the HTX crystallization facility (Proposal ID: BIOSTRUCTX_8167) and SOLEIL, ESRF and ALBA synchrotrons for provision of measurement time and thank their staff for help with data collection. The authors acknowledge the support of i3S Scientific Platforms (https://www.i3s.up.pt/scientific-platforms.php) Advanced Light Microscopy, member of the national infrastructure PPBI-Portuguese Platform of BioImaging (supported by POCI-01-0145-FEDER-022122), Animal Facility, Biochemical and Biophysical Technologies and X-ray Crystallography. A special thanks to Dr. Marc Graille and Dr. João Morais Cabral for constructive discussions in structural biology and Dr. Dimitri Panagiotis Papatheodorou for providing plasmid p327.

## Author contributions

J.L., A.d.V and N.M.S.S. conceived the study and the experiments; D.D. conceived, performed and analyzed SAXS experiments; R.B. conceived, performed and analyzed black lipid experiments; J.L. performed X-ray crystallography and solved the AIP56 structure with final validation by P.J.B.P.; J.L., C.P., R.D.P., I.S.R., L.M.G.P. and B.P. performed the biochemical and cell-based experiments; J.L., A.d.V. and N.M.S.S. analyzed the results with contributions from J.E.A. C.P. and P.O., performed statistical analysis. J.L., C.P. and N.M.S.S. prepared the figures; J.L. and N.M.S.S. wrote the manuscript with contribution by A.d.V., J.E.A. and P.J.B.P. All authors reviewed and approved the final version of the manuscript.

## Competing interests

The authors declare no competing interests.

## Additional information

[1]Fish Immunology and Vaccinology Group, IBMC-Instituto de Biologia Molecular e Celular, Universidade do Porto, 4200–135 Porto, Portugal. [2]Fish Immunology and Vaccinology Group, Instituto de Investigação e Inovação em Saúde, Universidade do Porto, 4200–135 Porto, Portugal. [3]Doctoral Program in Molecular and Cell Biology (MCbiology), Instituto de Ciências Biomédicas Abel Salazar - Universidade do Porto, Porto, Portugal. [4]EPIUnit, ICBAS-Instituto de Ciências Biomédicas Abel Salazar, Universidade do Porto, Porto, Portugal. [5]Biomolecular Structure Group, IBMC-Instituto de Biologia Molecular e Celular, Universidade do Porto, 4200–135 Porto, Portugal. [6]Macromolecular Structure Group, Instituto de Investigação e Inovação em Saúde, Universidade do Porto, 4200–135 Porto, Portugal. [7]ICBAS-Instituto de Ciências Biomédicas Abel Salazar, Universidade do Porto, Porto, Portugal. [8]Organelle Biogenesis and Function, IBMC-Instituto de Biologia Molecular e Celular, Universidade do Porto, 4200–135 Porto, Portugal. [9]Organelle Biogenesis and Function, Instituto de Investigação e Inovação em Saúde, Universidade do Porto, 4200–135 Porto, Portugal. [10]Université Paris-Saclay, CEA, CNRS, Institute for Integrative Biology of the Cell (I2BC), 91198 Gif-sur-Yvette, France. [11]Science Faculty, Constructor University, Bremen, Germany. ✉e-mail: johnny.lisboa@i3s.up.pt; nsantos@i3s.up.pt

