## [Peer Review File · Nature Communications]

Unconventional structure and mechanisms for membrane interaction and translocation of the NF- κ B-targeting toxin AIP56REVIEWER COMMENTS

Reviewer #1 (Remarks to the Author):

In this study, the authors solved the structure of AIP56. The authors characterize ability of the linker domain to promote translocation. In contrast to diphtheria toxin, the authors suggest additional residues outside of the linker between catalytic and receptor binding domain are needed for toxicity. Overall, the experiments are well performed, except the western blots seem variable. However, the advance is descriptive, and the wide applicability to other proteins is not demonstrated. Consequently, the results will have limited appeal to specialists in the field.

Major Points

1. The authors suggest many other proteins could use this strategy, but provide no evidence that they do.
2. The linker domain remains required, so the finding that additional residues are needed is oversold.
3. The discussion is speculative. Ideas listed in lines 422-425, or lines 431-435 should be tested to bolster the manuscript's impact.
4. The error bars on the western blots (Figs 4C and 5B) are large. Individual points should be displayed to better see the variation in results.

Reviewer #2 (Remarks to the Author):

Key results - The novel, pH-dependent pore-forming mechanism of AIP56 described in this manuscript involving all three domains of AIP56 was effectively supported by the biochemical, biophysical, structural, and computational work done during this investigation. Employing a hybrid molecular model that was experimentally-derived and computationally-derived then fit to SAXS data represents an influential form of structural biology.

Validity - Overall, the data interpretation and conclusions were carefully drawn from a nice array of good data sets.

Significance - The multifaceted structural analysis provided in this study should provide functional insight into many AB toxins that are AIP56-like ultimately contributing to the development of specific inhibitors of AIP56 and homologous toxins that impact many aspects of biology including human health.

Data and methodology - Overall, the scientific approach was excellent as was the data quality. The X-ray

and SAXS data were of good quality as were the molecular models derived from each data set. The biochemical assays demonstrating the middle and receptor-binding domains were required for pore formation were also well-done and persuasive. The site-directed mutagenesis assays strongly suggesting a pH-dependent conformational change is necessary for catalytic subunit delivery to the cytoplasm were also well-thought-out experiments providing straightforward conclusions.

Clarity and context - This very well-written manuscript clearly presented the key results of this investigation with ample support, and reference to, from previous studies.

Reviewer #3 (Remarks to the Author):

This study by Lisboa et al. investigates the structure-function relationship of the bacterial AB toxin AIP56. Using AI structure prediction, macromolecular crystallography, and small-angle X-ray scattering, the authors reveal the three-domain organization of AIP56, highlighting the unique middle domain that connects the receptor binding and catalytic domains. Unlike other AB toxins, AIP56's middle domain is relatively small and exhibits a simple structure.

Through various assays, Lisboa et al. demonstrate that structural elements involved in translocation are dispersed across the other two domains, leading to two key observations:

- 1) both the middle and receptor-binding domains are essential for pore formation
- 2) pH-sensing residues controlling conformational changes are located in the carboxyl-terminal portion of the catalytic domain.

The study suggests an exclusive evolutionary origin of the AIP56 middle domain and its homologous toxins.

These findings have significant implications that extend beyond their immediate scope. Similar toxins possessing predicted three-dimensional structures and homologous domains could be susceptible to targeted inhibition, thereby potentially influencing aquaculture, agriculture, and human health. This consideration encompasses putative homologous toxins known as AIP56-like toxins in bacteria, as well as putative toxins possessing domains homologous to the catalytic or receptor-binding domain of AIP56 (referred to as AIP56-related toxins) found in bacteriophages, bacteria, and insects.

The manuscript is very clearly written and structured, making it accessible to readers who are not familiar with this specific topic. The scientific question being addressed is presented in a manner that is easy to comprehend, allowing a broad audience to understand the research objectives.

The experimental procedures are in general well explained, enabling readers to follow the methodology employed. The results are presented in a clear and organized manner, facilitating the interpretation of the findings. The conclusions drawn from the results are sound and logically derived, enhancing the overall coherence of the study.

Exemplary, the authors have made the SAXS and MX data openly accessible in the respective databases.

Based on this, I recommend the article for publication.

However, to further enhance the overall quality of the work, I would suggest addressing a few additional points.

2 or 3 domain predictions:

- It is not quite clear if a 2domain --- or 3 domain – or 2 domain ‘including small somewhat structured middle domain was predicted.
- In line Introduction 48, says only 2 domains were predicted, however, the alpha fold advance model seemed to be suitable for molecular replacement. (were there any issues, other linkers are missing in the electron density map, however, the ‘unstructured’ middle domain, seems to be seen in the crystal structure, what are the B-factors like of this linker region
- At least a few of the predictions shown in FigS2b also show some structural arrangement of the middle domain (perhaps quantify somewhere in text).

SAXS data:

- Mention the good agreement between SAXS derived MW estimates and expected MW of monomer (see SAXS table). From Porod volume 57.5 kD, From consensus Bayesian assessment 55.6 ± 6 , Calculated monomeric MM from sequence 57.25
- Note, the color in the legend of the SAXS data is swapped (Fig 1 b + c). The red lines are the fits, the grey curves the experimental data.
- Perhaps it should be mentioned that without the missing linkers (just chain A from r crystal structure – fit is quite bad)
- It is stated ‘address the interdomain conformational flexibility of AIP56 with SREFLEX -* but no further interpretation of the NMA result is given, except for the overlay in Figure 1C If aligning the Sreflex (violet) and the modeller structure (light blue) so that residues 1-333 (grey) overlap, one can see that the divergence of the structure is around residue 305

Thus to test the flexibility – EOM (Tria, G., Mertens, H. D. T., Kachala, M. & Svergun, D. I. (2015) Advanced ensemble modeling of flexible macromolecules using X-ray solution scattering. IUCrJ 2, 207-217 DOI) approach was applied (zip file) attached. The NTERM domain and Cterm domain were created by removing residues 306-315 -* 10.000 random models generated with the 10 residue linker -* from this pool a subset is selected that as an ensemble fits the data. From this subset, structural parameters such as RG and DMAX are compared with the parameters of the large pool. Here, the selection, suggests slightly flexible with the preference of

domains quite close together (no selection of models with the extended linker). This analysis is in line with the rest of the study and addresses flexibility issue more than just NMA

The resulting fit is similar to SREFLEX with χ^2 of 1.54

- For Further studies (outside of the scope here) it would of course be interesting to measure at different pH and perhaps see a shift in oligomeric state. Perhaps this could be discussed.
- Line 554, add the definition of momentum transfer, as used here
- Were there really interparticle effects seen after the separation of 13.4 mg/ml sample (perhaps different buffer regions would result in a curve with linear Guinier regions (which range of the lower
- How were the frames normalized before displayed on absolute scale? (transmitted beam?)
- In light of the possible oligomerization process, it might be good to display the elution profile as supplementary figure and show the stable RG estimates across the peak.

Minor changes

- Line 120: Mention use of molecular crystallography of means for structure determination in the introduction

- Here, SAXS is used to confirm monomeric structure in solution. In the pdb entry 7zpf.pdb it is assigned as a dimer. Would there be any further implications if dimerization occurred (for example under different conformational conditions (eg. SAXS is measured at quite high salt). Where SAXS measurements performed in batch as well, that could give insights into the oligomerization state. What does the SAXS elution profile look like? A hint of dimeric (or even tetrameric) components that were separated from the sample?
- Line 204 – mBMMDM is first/only defined in Figure legend 3 / Methods Section 4
- Paragraph starting with line 200 could benefit from re-structuring.
For example:

Many short-trip single-chain toxins typically exhibit a three-domain organization, where the middle domain is dedicated to pore formation and facilitating the translocation of the catalytic domain into the cytosol. This raises the question of whether AIP56's middle domain, despite its structural simplicity, could be responsible for pore formation. First evidence in favor of this concept was collected in previous studies that investigated the successful delivery of the model protein β -lactamase (Bla) into the cytosol of mouse bone marrow-derived macrophages (mBMMDM). This was achieved by studying a chimera of Bla fused to the middle and receptor-binding domains of AIP56.

Building upon these findings, we

- Line 213: quantify strong acidification (pH 4.8 – 5.0)

Reviewer #4 (Remarks to the Author):

I have read with pleasure the manuscript submitted by Lisboa and colleagues titled “Structural and functional characterization of the NF- κ B-targeting toxin AIP56 from *Photobacterium damsela* subsp. *piscicida* reveals a novel mechanism for membrane interaction and translocation”. The work is of great interest to comprehend the interactions between bacterial toxins and membranes which are essential for their internalization into cells as well as the mechanisms that lead to apoptosis.

The structure of the AIP56 protein has been resolved by means of SAXS experiments and bioinformatic analysis. It was evidenced the protein possesses a catalytic, a middle and a receptor domain. Certain AA residues of the catalytic domain were found to be crucial in a pH-dependent conformational change of the protein which appears necessary for membrane translocation. The presented data is sound and the manuscript clear, well written and in the scope of Nature Communications. I recommend publication after the following minor point is addressed concerning the interaction of the protein with black lipid membranes.

– The final protein concentration for the BLM experiments could be indicated on all figures, similarly to Fig. 2B.

– In Fig. 2, it is suggested that among the truncated proteins, only the AIP56L258-N497 presented a

slight membrane activity with a very high concentration of protein (800 nM). The native AIP56 and the BLA chimera were tested at only 14 nM and presented an important membrane activity. In Fig. 3E and 5C, the mutants were also tested at 14 nM and some of them presented an important activity. Does AIP56L258-N497 present a membrane activity at a concentration of 14 nM? If not, the conclusion regarding the necessity of the receptor domain for membrane interaction may not be completely accurate.

Point by point response to reviewer's comments

We thank the reviewers for their comments and insightful suggestions. We addressed all points raised, modified the manuscript and added new figures, accordingly. We hope that the reviewers find the revised version of our manuscript suitable for publication.

Below, please find a point-by point reply to the reviewer's comments.

Please note that, unless otherwise stated, the line numbers included in the responses to the reviewers' comments correspond to the line numbers in the manuscript word file with track changes in "All Markup" view.

REVIEWER COMMENTS

Reviewer #1 (Remarks to the Author):

In this study, the authors solved the structure of AIP56. The authors characterize ability of the linker domain to promote translocation. In contrast to diphtheria toxin, the authors suggest additional residues outside of the linker between catalytic and receptor binding domain are needed for toxicity. Overall, the experiments are well performed, except the western blots seem variable. However, the advance is descriptive, and the wide applicability to other proteins is not demonstrated. Consequently, the results will have limited appeal to specialists in the field.

R: We thank the reviewer for the analysis and comments.

Major Points

1. The authors suggest many other proteins could use this strategy, but provide no evidence that they do.

R: Although this manuscript focuses on AIP56, we could not avoid drawing attention to the eventual structural and functional similarities between AIP56 and AIP56-like toxins or AIP56-related toxins. Our suggestion that there are similarities between these proteins is based on: (i) AlphaFold predictions of a three domain structure similar to that of AIP56 for AIP56-like toxins; and (ii) AIP56-like toxins conserves the elements that in AIP56 are involved in the translocation process, such as pH-sensing residues in a helical hairpin in the C-terminal region of the catalytic domain (Supplementary Fig. 7d) and a set of conserved aspartate residues at the tip of the linker (Supplementary Fig. 2).

We agree with the reviewer on the importance of carrying out studies to characterize structurally and functionally AIP56-like (and AIP56-related) toxins. The number of these putative toxins is continuously increasing, and therefore disclosing their biological roles and mechanisms of action may have significant impacts in aquaculture, agriculture and human health. However, the detailed characterization of these toxins is a challenging endeavor that will have to be necessarily carried out (by us or others) outside of the scope of the present work, which focuses on the peculiar structural organization of AIP56. We believe that this work stands by itself and is of great interest for the community, as it is the first description of a single-chain AB toxin in which the elements controlling pore formation and translocation are scattered throughout its domains.

2. The linker domain remains required, so the finding that additional residues are needed is oversold.

R: Please note that we have shown that replacing pH-sensing residues in the catalytic domain led to uncontrolled toxin unfolding, abolished the interaction with the membrane and the consequent intoxication process. Thus, in our opinion, the fact that the middle domain is required does not diminish the importance of showing that specific pH-sensing residues of the catalytic domain control the timing of pore formation. In fact, this is an essential step of intoxication, as there must be concertation between toxin unfolding and pore formation, in order to ensure that they occur at the appropriate place and time to allow translocation of the catalytic domain from acidified endosomes

to the cytosol. One of the novelties of this work is showing that, in AIP56, it is the catalytic domain (and not a “typical” translocation domain) that controls its own translocation.

3. The discussion is speculative. Ideas listed in lines 422-425, or lines 431-435 should be tested to bolster the manuscript's impact.

R: We agree that the ideas listed in lines 427-429 (previously 422-425) and 436-440 (previously 431-435) are very much worth exploring, but they stand alone as an independent work, out of the immediate scope of the present work. Nevertheless, we decided to include those ideas (and others) throughout the discussion so that they could prompt future studies. We made a choice of focusing mainly on fundamental ideas derived from this study, mechanistic hypotheses, and potential implications that this study may have beyond AIP56, in order to make the discussion more appealing for a broader readership, while describing, contextualizing and specifically discuss the data obtained in the Results section.

4. The error bars on the western blots (Figs 4C and 5B) are large. Individual points should be displayed to better see the variation in results.

R: We agree with the reviewer. Individual data points are now displayed in the plots, as suggested, and raw data is also available as Source Data.

Reviewer #2 (Remarks to the Author):

Key results - The novel, pH-dependent pore-forming mechanism of AIP56 described in this manuscript involving all three domains of AIP56 was effectively supported by the biochemical, biophysical, structural, and computational work done during this investigation. Employing a hybrid molecular model that was experimentally-derived and computationally-derived then fit to SAXS data represents an influential form of structural biology.

Validity - Overall, the data interpretation and conclusions were carefully drawn from a nice array of good data sets.

Significance - The multifaceted structural analysis provided in this study should provide functional insight into many AB toxins that are AIP56-like ultimately contributing to the development of specific inhibitors of AIP56 and homologous toxins that impact many aspects of biology including human health.

Data and methodology - Overall, the scientific approach was excellent as was the data quality. The X-ray and SAXS data were of good quality as were the molecular models derived from each data set. The biochemical assays demonstrating the middle and receptor-binding domains were required for pore formation were also well-done and persuasive. The site-directed mutagenesis assays strongly suggesting a pH-dependent conformational change is necessary for catalytic subunit delivery to the cytoplasm were also well-thought-out experiments providing straightforward conclusions.

Clarity and context - This very well-written manuscript clearly presented the key results of this investigation with ample support, and reference to, from previous studies.

R: We thank the reviewer for the thorough analysis of our work and for the very positive words.

Reviewer #3 (Remarks to the Author):

*there are figures included, thus, better look at the attached pdf file. there is also a *zip file containing additional SAXS analysis.

REVIEW:

This study by Lisboa et al. investigates the structure-function relationship of molecular crystallography, and small-angle X-ray scattering, the authors reveal the three-domain organization of AIP56, highlighting its unique middle domain that connects the receptor binding and catalytic domains. Unlike other AB toxins, AIP56's middle domain is relatively small and exhibits a simple structure.

Through various assays, Lisboa et al. demonstrate that structural elements involved in translocation are dispersed across the other two domains, leading to two key observations:

- 1) both the middle and receptor-binding domains are essential for pore formation
- 2) pH-sensing residues controlling conformational changes are located in the carboxyl-terminal portion of the catalytic domain.

The study suggests an exclusive evolutionary origin of the AIP56 middle domain and its homologous toxins. These findings have significant implications that extend beyond their immediate scope. Similar toxins possessing predicted three-dimensional structures and homologous domains could be susceptible to targeted inhibition, thereby potentially influencing aquaculture, agriculture, and human health. This consideration encompasses putative homologous toxins known as AIP56-like toxins in bacteria, as well as putative toxins possessing domains homologous to the catalytic or receptor-binding domain of AIP56 (referred to as AIP56-related toxins) found in bacteriophages, bacteria, and insects.

The manuscript is very clearly written and structured, making it accessible to readers who are not familiar with this specific topic. The scientific question being addressed is presented in a manner that is easy to comprehend, allowing a broad audience to understand the research objectives.

The experimental procedures are in general well explained, enabling readers to follow the methodology employed. The results are presented in a clear and organized manner, facilitating the interpretation of the findings. The conclusions drawn from the results are sound and logically derived, enhancing the overall coherence of the study. Exemplary, the authors have made the SAXS and MX data openly accessible in the respective databases. Based on this, I recommend the article for publication.

R: We thank the reviewer for the thorough analysis of our work and for the positive words.

However, to further enhance the overall quality of the work, I would suggest addressing a few **additional points**.

2 or 3 domain predictions:

- It is not quite clear if a 2domain --- or 3 domain – or 2 domain ‘including a small somewhat structured middle domain was predicted.

R: There is indeed some confusion in our text when referring to the domain structure of AIP56. On one hand, we summarize previous data from our laboratory (based on Blast analyses and limited proteolysis assays) suggesting that AIP56 comprised two domains. On the other hand, the data presented in this manuscript shows that AIP56 has a 3-domain structure. The transitions in the text from previous data to the new results were not always clear and therefore created some confusion. To improve this aspect, we changed the sentences in the abstract (line 48-51) and introduction (lines 101-106) to increase clarity.

Sentence in Abstract (lines 48-51)

“In this work, the determination of the three-dimensional structure of AIP56 showed that, instead of a two-domain organization suggested by previous studies, AIP56 has three-domains: a NleC-like catalytic domain associated with a small middle domain that contains the linker-peptide, followed by the receptor-binding domain.”

Sentence in Introduction (lines 101-106)

“In previous studies¹⁸, amino acid sequence and limited proteolysis analyses suggested a two-domain organization for AIP56: (i) an A domain homologous to NleC (non-LEE encoded effector C), a type III effector with enzymatic activity towards NF-kB p65 that is injected into the cytosol by the type III secretion system of several enteric bacteria associated with human diseases²⁴⁻²⁶; and (ii) a receptor-binding B domain, homologous to Protein D from bacteriophage APSE2 (*Acyrtosiphon pisum* secondary endosymbiont 2)²⁷⁻³⁰.”

• In line Introduction 48, says only 2 domains were predicted, however, the alpha fold advance model seemed to be suitable for molecular replacement. (were there any issues, other linkers are missing in the electron density map, however, the 'unstructured' middle domain, seems to be seen in the crystal structure, what are the B-factors like of this linker region

R: In addition to the question of the predicted 2 domains, answered above, the reviewer is correct in that the middle domain is visible in all 4 copies of AIP56 in the asymmetric unit. This can be explained by the contacts established between the middle domain and the catalytic and the receptor-binding domains, stabilizing this linker region in the crystal (Fig. 1f). The average B-factors are as follow:

- Chain A: middle domain= 66.2 Å², whole molecule= 57.4 Å²
- Chain B: middle domain= 72.2 Å², whole molecule= 57.4 Å²
- Chain C: middle domain= 103.7 Å², whole molecule= 79.5 Å²
- Chain D: middle domain= 102.1 Å², whole molecule= 70.1 Å²

The image below shows the B-factor distribution on the crystal structure of the most complete chain (chain A).

• At least a few of the predictions shown in FigS2b also show some structural arrangement of the middle domain (perhaps quantify somewhere in the text).

R: We assume the reviewer is referring to the AlphaFold generated models shown in Fig. S1b (now Supplementary Fig. 1c).

The reviewer is right regarding the apparent structural rearrangement of the middle domain, more evident in *A. nasoniae* toxins, which present lower amino acid conservation compared to the other AIP56-like toxin (Supplementary Fig. 2b). However, we must keep in mind that this intrinsically flexible segment is the least reliable region of each of the AlphaFold models and, in our view, trying to quantify their apparent structural differences will add very little (if any) information. This is the main reason why we opted for a more visual depiction of those differences. In order to underscore this, we added to the figure (now Supplementary Fig. 1c) the analysis confirming that there is a strong structural similarity between the catalytic and receptor-binding domains (well predicted by AlphaFold), and rearranged the text to better reflect the structural differences observed in the middle domain.

Lines 149-157

“The data revealed highly similar structures for all proteins analyzed. Major structural differences were observed only in the middle domains, particularly in the linker region (Supplementary Fig. 1c). Moreover, despite the high aminoacidic (Supplementary Fig. 2b) and structural (Supplementary Fig. 1c) conservation of the catalytic and receptor-binding domains, the AlphaFold-generated structures for *A. nasoniae* toxins resulted in distinct relative positions of those domains. Whether this results from inaccurate predictions of the middle domains by

AlphaFold2 or corresponds to the real positions of the catalytic and receptor-binding domains in *A. nasoniae* toxins remains to be investigated.”

SAXS data:

- Mention the good agreement between SAXS-derived MW estimates and the expected MW of the monomer (see SAXS table). From Porod volume 57.5 kD, From consensus Bayesian assessment 55.6 ± 6 , Calculated monomeric MM from sequence 57.25

R: As requested, we have added a sentence (lines 141-143) mentioning the good agreement between the sequence-derived molecular mass and experimental estimates.

“The monomeric nature of AIP56 in solution is confirmed by the remarkable agreement between the sequence-derived molecular mass and experimental estimates (Supplementary Table 1 and Supplementary Fig. 1b).”

- Note, the color in the legend of the SAXS data is swapped (Fig 1 b + c). The red lines are the fits, and the grey curves the experimental data.

R: Thank you for pointing this out. The legend of the figure has been corrected.

- Perhaps it should be mentioned that without the missing linkers (just chain A from r crystal structure – fit is quite bad)

R: We thank the reviewer for this suggestion. We have added that information to the text (lines 137-141).

“SAXS analysis of AIP56 (Supplementary Table 1 and Supplementary Fig. 1b) revealed a reasonable fit between the experimental scattering data and the theoretical scattering profile calculated with CRY SOL³⁷ for the structural model of monomeric AIP56 obtained with Modeller ($\chi^2 = 5.05$) (Fig. 1b, upper panel). The fit becomes quite poor when using the crystal structure of AIP56 without the missing linkers (ND1-4, $\chi^2 = 39$).”

- It is stated ‘address the interdomain conformational flexibility of AIP56 with SREFLEX \diamond but no further interpretation of the NMA result is given, except for the overlay in Figure. 1C. If aligning the Sreflex (violet) and the modeller structure (light blue) so that residues 1-333 (grey) overlap, one can see that the divergence of the structure is around residue 305. Thus to test the flexibility – EOM (Tria, G., Mertens, H. D. T., Kachala, M. & Svergun, D. I. (2015) Advanced ensemble modeling of flexible macromolecules using X-ray solution scattering. IUCrJ 2, 207-217 DOI) approach was applied (zip file) attached. The NTERM domain and Cterm domain were created by removing residues 306-315 \diamond 10.000 random models generated with the 10 residue linker \diamond from this pool a subset is selected that as an ensemble fits the data. From this subset, structural parameters such as RG and DMAX are compared with the parameters of the large pool. Here, the selection, suggests slightly flexible with the preference of domains quite close together (no selection of models with the extended linker). This analysis is in line with the rest of the study and addresses the flexibility issue more than just NMA

The resulting fit is similar to SREFLEX with chi2 of 1.54

- see PDF File with Figures

R: We greatly appreciate the analysis performed by the reviewer and are glad to see that different approaches led to similar conclusions.

The rationale behind the use of SREFLEX was as follows: when calculating the scattering pattern of the crystal structure completed by Modeller, we were impressed by the global agreement with the experimental data, suggesting that the structure in solution was very similar to the completed structure. As clearly seen in the distribution of reduced residuals, differences were mainly located in the small angle region, suggesting a slightly different arrangement of domains. We thus resorted to the Normal Mode Analysis as implemented in SREFLEX, an approach well-suited to optimize the global conformation in the vicinity of the starting model. The scattering

curve of the resulting model exhibited an excellent agreement with experimental data while the corresponding movement from the starting structure was of limited amplitude (when superimposing the N-term domains, rms 320-497 = 1.9 Å). The result is suggestive of restricted flexibility in solution. The EOM analysis suggested and actually performed by the reviewer is interesting but its results, as stated by the reviewer, while supporting our view of a restricted flexibility, do not bring much additional information. Accordingly, we prefer not to add the EOM analysis to an already complex article in which SAXS data are not central, especially considering that the reviewer's analysis will be available for readers to consult, as we will allow publishing the peer review history.

- For Further studies (outside of the scope here) it would of course be interesting to measure at different pH and perhaps see a shift in oligomeric state. Perhaps this could be discussed.

R: We agree with the reviewer that this is an interesting point to address outside this work and that we are interested in exploring. However, we chose not to include a discussion around this point in the manuscript because in the absence of experimental data, it would be too speculative.

- Line 554, add the definition of momentum transfer, as used here

R: We have added the definition as recommended by the reviewer (lines 552-554).

“The sample-to-detector (Dectris Eiger 4M) distance was set to 2000 mm and the wavelength λ to 1.0 Å, allowing useful data collection over the momentum transfer range of $0.005 \text{ \AA}^{-1} < q < 0.5 \text{ \AA}^{-1}$ ($q=4\pi \sin(\theta)/\lambda$).”

- Were there really interparticle effects seen after the separation of 13.4 mg/ml sample (perhaps different buffer regions would result in a curve with linear Guinier regions (which range of the lower

R: Yes, attractive interactions were apparent in the 13.4 mg/ml sample, so that a merge operation was performed between the low concentration data (initial concentration of 3.2 mg/ml) at small angle, most sensitive to interactions and the higher concentration at higher angle to provide better statistics in the outer range of the curve.

How were the frames normalized before being displayed on absolute scale? (transmitted beam?)

R: As mentioned in the text (lines 559-560), data were put on absolute scale in cm^{-1} using water scattering.

“The scattered intensities were displayed on an absolute scale using the scattering by water.”

- In light of the possible oligomerization process, it might be good to display the elution profile as supplementary figure and show the stable RG estimates across the peak.

R: The elution profile has been added as Supplementary Fig. 1b.

Minor changes

- Line 120: Mention use of molecular crystallography of means for structure determination in the introduction

R: The reference to crystallography has been added as suggested.

Lines 117-119

“In this work, the three-dimensional structure of AIP56 was determined using X-ray crystallography and the elements involved in acidic pH-induced conformational changes, membrane interaction and translocation were characterized.”

- Here, SAXS is used to confirm monomeric structure in solution. In the pdb entry 7zpf.pdb it is assigned as a dimer. Would there be any further implications if dimerization occurred (for example under different conformational

conditions (eg. SAXS is measured at quite high salt). Where SAXS measurements performed in batch as well, that could give insights into the oligomerization state. What does the SAXS elution profile look like? A hint of dimeric (or even tetrameric) components that were separated from the sample?

R: The PDB erroneously assigned the oligomeric state of entry 7zpf to dimeric, due to a crystallization artifact, where coordination of a nickel ion by the histidine-tags of two AIP56 molecules originates an artificial dimer (see figure below). A correction has been requested and the entry has been meanwhile updated. However, there is a hint of dimeric component visible on the UV trace of the SAXS elution profile that has been added to the supplementary information (see added Supplementary Fig. 1b).

- Line 204 – mBMDM is first/only defined in Figure legend 3 / Methods Section 4

R: Thank you for noting this error. This has been corrected (lines 207-212).

“The structural organization of AIP56 together with previous results showing that a chimera comprising β -lactamase (Bla) fused to the middle and receptor-binding domains of AIP56 (Bla^{L19-W286}AIP56^{L258-N497}) (Fig. 2a) was able to deliver Bla into the cytosol of mouse bone marrow derived macrophages (mBMDM)²³ support this concept.”

- Paragraph starting with line 200 could benefit from re-structuring.

For example: Many short-trip single-chain toxins typically exhibit a three-domain organization, where the middle domain is dedicated to pore formation and facilitating the translocation of the catalytic domain into the cytosol. This raises the question of whether AIP56's middle domain, despite its structural simplicity, could be responsible for pore formation. First evidence in favor of this concept was collected in previous studies that investigated the successful delivery of the model protein β -lactamase (Bla) into the cytosol of mouse bone marrow-derived macrophages (mBMDM). This was achieved by studying a chimera of Bla fused to the middle and receptor-binding domains of AIP56. Building upon these findings, we.....

R: We thank the reviewer for the suggestion. The paragraph has been rearranged as follows:

Lines 204-213

“Many short-trip single-chain toxins exhibit a three-domain organization, where the middle domain is dedicated to pore formation and facilitates the translocation of the catalytic domain into the cytosol^{2,3}. This raises the question of whether AIP56's middle domain, despite its structural simplicity, could be responsible for pore formation. The

structural organization of AIP56 together with previous results showing that a chimera comprising β -lactamase (Bla) fused to the middle and receptor-binding domains of AIP56 (Bla^{L19-W286}AIP56^{L258-N497}) (Fig. 2a) was able to deliver Bla into the cytosol of mouse bone marrow derived macrophages (mBMDM)²³ support this concept. To further test this, several truncated versions of AP56 were used in experiments with black lipid bilayers (Fig. 2a), namely:....”.

- Line 213: quantify strong acidification (pH 4.8 – 5.0)

R: The pH values (pH 4.8 – 5.0) were added after “acidification” (Lines 215-219).

“Of these, only the construct containing both the middle and receptor-binding domains (AIP56^{L258-N497}) displayed membrane-interacting activity after acidification (pH 4.8-5.0) of the cis-side of the membrane, although the observed activity was different and much lower than that obtained with intact AIP56 (Fig. 2b).”

Reviewer #4 (Remarks to the Author):

I have read with pleasure the manuscript submitted by Lisboa and colleagues titled “Structural and functional characterization of the NF- κ B-targeting toxin AIP56 from *Photobacterium damsela* subsp. piscicida reveals a novel mechanism for membrane interaction and translocation”. The work is of great interest to comprehend the interactions between bacterial toxins and membranes which are essential for their internalization into cells as well as the mechanisms that lead to apoptosis.

The structure of the AIP56 protein has been resolved by means of SAXS experiments and bioinformatic analysis. It was evidenced the protein possesses a catalytic, a middle and a receptor domain. Certain AA residues of the catalytic domain were found to be crucial in a pH-dependent conformational change of the protein which appears necessary for membrane translocation. The presented data is sound and the manuscript clear, well written and in the scope of Nature Communications. I recommend publication after the following minor point is addressed concerning the interaction of the protein with black lipid membranes.

- The final protein concentration for the BLM experiments could be indicated on all figures, similarly to Fig. 2B.

R: The protein concentration is now indicated in the corresponding figures, as suggested.

- In Fig. 2, it is suggested that among the truncated proteins, only the AIP56^{L258-N497} presented a slight membrane activity with a very high concentration of protein (800 nM). The native AIP56 and the BLA chimera were tested at only 14 nM and presented an important membrane activity. In Fig. 3E and 5C, the mutants were also tested at 14 nM and some of them presented an important activity. Does AIP56^{L258-N497} present a membrane activity at a concentration of 14 nM? If not, the conclusion regarding the necessity of the receptor domain for membrane interaction may not be completely accurate.

R: Experiments with truncated constructs were not conducted at a concentration of 14 nM. However, AIP56^{L258-N497} has been tested at 300 nM and the activity was similar to the one observed at 800 nM (this information was now added to the legend of Fig. 2b). Also, please note that all truncated constructs and chimeras were tested at 800 nM and: (i) only the proteins containing both the middle and receptor-binding domain (AIP56, AIP56^{L258-N497} and Bla^{L19-W286}AIP56^{L258-N497}) displayed membrane activity; (ii) no activity was observed for the construct containing the catalytic domain plus middle domain (AIP56^{N1-E307}); and (iii) no activity was detected for the chimera containing AIP56 catalytic and middle domains fused to diphtheria receptor-binding domain (AIP56^{N1-E307}DTR^{S406-S560}). Therefore, our interpretation that pore formation requires the receptor-binding domain was not only based on the result obtained with AIP56^{L258-N497}.

For clarity, we have also rearranged the description of these results in point “Pore formation requires both the middle and receptor-binding domains” (previous point 2.3 of the Results section) as follows (lines 204-228):

“Many short-trip single-chain toxins exhibit a three-domain organization, where the middle domain is dedicated to pore formation and facilitates the translocation of the catalytic domain into the cytosol^{2,3}. This raises the question of whether the middle domain of AIP56, despite its structural simplicity, could mediate pore formation. The structural organization of AIP56 together with previous results showing that a chimera comprising β -lactamase (Bla) fused to the middle and receptor-binding domains of AIP56 (Bla^{L19-W286}AIP56^{L258-N497}) (Fig. 2a) was able to

deliver Bla into the cytosol of mouse bone marrow derived macrophages (mBMDM)²³ support this concept. To further test this, several truncated versions of AIP56 were used in experiments with black lipid bilayers (Fig. 2a), namely: the catalytic domain alone (AIP56^{N1-G256}), the catalytic domain with the middle domain (AIP56^{N1-E307}), the middle domain with the receptor-binding domain (AIP56^{L258-N497}) and the receptor-binding domain alone (AIP56^{T299-N497}). Of these, only the construct containing both the middle and receptor-binding domains (AIP56^{L258-N497}) displayed membrane-interacting activity after acidification (pH 4.8-5.0) of the cis-side of the membrane, although the observed activity was different and much lower than that obtained with intact AIP56 (Fig. 2b). In agreement with this, the Bla^{L19-V286}AIP56^{L258-N497} chimera interacted with black lipid membranes, whereas a chimera in which the AIP56 receptor-binding domain was replaced by the DT receptor-binding (DTR) domain (AIP56^{N1-E307}DTR^{S406-S560}) did not display membrane activity (Fig. 2b) and was unable to deliver the AIP56 catalytic domain into the cytosol of U-2 OS cells (Supplementary Fig. 4a,b). Altogether, the black lipid bilayer experiments suggest that the interaction of AIP56 with the membrane requires both the middle and receptor-binding domains.”

REVIEWER COMMENTS

Reviewer #1 (Remarks to the Author):

In the revised version, the authors failed to address most of my concerns.

1) If the authors do not plan to test other proteins that might use this strategy, at most mention of other proteins should get 1 line in the discussion, and should be removed from the abstract.

2) Authors' assertions about the need for additional residues remains oversold, so these statements should be tempered.

3) Lines 414-427 in the discussion remain speculative. Absent experimental evidence, this should be condensed to 1 or 2 lines about future directions maximum.

Reviewer #3 (Remarks to the Author):

The authors have adequately addressed all of my comments and incorporated corresponding revisions into the manuscript.

I endorse the publication of this article.

Reviewer #4 (Remarks to the Author):

I read the manuscript resubmitted by Lisboa and colleagues. The structure of the bacterial protein AIP56 was resolved using an approach bioinformatics and SAXS experiments. It was evidenced that the toxin induces pore formation induced via its middle and receptor binding domains. This phenomenon was shown to be pH-dependent since acidification triggered conformational changes that favored interactions with lipid membranes. Authors provided qualitative data supporting their interpretations and conclusions.

The article is clear and well-written and the authors addressed all questions and remarks raised by all reviewers.

I recommend publication of the article in its present form.

Point by point response to reviewer's comments

We thank the reviewer for their questions and suggestions, and provide below (in blue font) a detailed description of the modifications made to the manuscript.

REVIEWER COMMENTS

Reviewer #1 (Remarks to the Author):

In the revised version, the authors failed to address most of my concerns.

- 1) If the authors do not plan to test other proteins that might use this strategy, at most mention of other proteins should get 1 line in the discussion, and should be removed from the abstract.

As requested, we removed the mention to the implications of this study on AIP56-like and AIP56-related toxins from the abstract (lines 55-57) and reduced it to one sentence in the final paragraph of the discussion (lines 420-421).

Also, in the sentence "Taken together, these results suggest that the hairpin within the catalytic domain of AIP56 contains pH-sensing residues that control the conformational changes necessary for pore formation by the middle and receptor-binding domains, a mechanism most likely conserved in AIP56-like toxins." (lines 302-304), "most likely" has been replaced by "that may be".

- 2) Authors' assertions about the need for additional residues remains oversold, so these statements should be tempered.

As suggested, sentences on "the need for additional residues" in the translocation process were toned down or removed from the manuscript.

The sentence "These results strongly suggest that the $\alpha 8$ - $\alpha 9$ hairpin is important for the translocation process." (lines 254-255) was rephrased by removing the word "strongly".

The sentence "While PFTs and short-trip AB toxins have evolved a variety of regulatory mechanisms to prevent premature pore formation^{53,84,85}, the mechanism here described for AIP56, where the pH-sensing hairpin of the catalytic domain acts as a molecular switch for timing pore formation and controlling its own translocation, is remarkable." (lines 371-374) has been deleted.

- 3) Lines 414-427 in the discussion remain speculative. Absent experimental evidence, this should be condensed to 1 or 2 lines about future directions maximum.

The final section of the Discussion (lines 415-432), reading:

"Finally, it is possible that in the process of translocation to the cytosol, and before or after its release upon cleavage by a protease, the catalytic domain would interact with Hsp90 through the region of its pH-sensing hairpin, which would assist its refolding^{19,23}.

The existence of putative homologous toxins with a predicted three-dimensional structure similar to that of AIP56 (AIP56-like toxins) in bacteria that have host organisms as evolutionarily distant as insects and humans, suggests that they may be functionally similar to AIP56 and also play a key role in virulence. In addition, there are putative toxins with domains homologous to the catalytic or receptor-binding domain of AIP56 (AIP56-related toxins) in bacteriophages (e.g., APSE2 and APSE7), bacteria (e.g., *V. nigrispulchritudo* and *Enterovibrio norvegicus*) and insects (e.g., *Danaus plexippus*, *Drosophila bipectinata* and *D. ananassae*³³). Overall, in addition to the implications it will have for the understanding of the intoxication mechanism of AIP56 and virulence of *Phdp*, the characterization of the structure-function relationship of AIP56 will have wider

implications and may contribute to the development of specific inhibitors of AIP56 and homologous toxins, with implications for aquaculture, agriculture and, eventually, human health.”,

has been drastically reduced and now reads:

“Finally, it is possible that, upon translocating to the cytosol, the catalytic domain interacts with Hsp90 through the region of its pH-sensing hairpin^{19,23}.

In summary, the structural and functional data described in this work provide new insights on the AIP56 translocation mechanism and may contribute to develop prophylaxis and treatments based on AIP56. Future work may elucidate whether the conclusions drawn for AIP56 can be extended to the increasing number of AIP56-like and -related toxins.”